

# Carbon dioxide retrieval from OCO-2 satellite observations using the RemoTeC algorithm and validation with TCCON measurements

Lianghai Wu[1], Otto Hasekamp[1], Haili Hu[1], Jochen Landgraf[1], Andre Butz[2,3], Joost aan de Brugh[1], Ilse Aben[1], Dave F. Pollard[4], David W. T. Griffith[5], Dietrich G. Feist[6], Dmitry Koshelev[7], Frank Hase[8], Geoffrey C. Toon[9], Hirofumi Ohyama[10], Isamu Morino[10], Justus Notholt [11], Kei Shiomi[12], Laura Iraci[13], Matthias Schneider[15], Martine de Maziére [14], Ralf Sussmann[15], Rigel Kivi[16], Thorsten Warneke[11], Tae-Young GOO[17], and Yao Té[7]

[1]SRON Netherlands Institute for Space Research, Utrecht, The Netherlands
[2]Institute of Atmospheric Physics, Deutsches Zentrum für Luft- und Raumfahrt e.V. (DLR), Wessling-Oberpfaffenhofen, Germany
[3]Meteorologisches Institut, Ludwig-Maximilians-Universität (LMU), Munich, Germany
[4]National Institute of Water and Atmospheric Research Ltd (NIWA), Lauder, New Zealand
[5]University of Wollongong, Wollongong, Australia
[6]Max Planck Institute for Biogeochemistry, Jena, Germany
[7]LERMA-IPSL, Sorbonne Universités, UPMC Univ Paris 06, CNRS, Observatoire de Paris, PSL Research University, 75005, Paris, France
[8]Karlsruhe Institute of Technology (KIT), IMK-ASF, Karlsruhe, Germany
[9]Jet Propulsion Laboratory, California Institute of Technology, Pasadena, California, USA
[10]National Institute for Environmental Studies (NIES), Tsukuba, Japan
[11]University of Bremen, Bremen, Germany
[12]Japan Aerospace Exploration Agency, Tsukuba, Japan
[13]NASA Ames Research Center, Moffett Field, CA, USA
[14]Royal Belgian Institute for Space Aeronomy, Brussels, Belgium
[15]Karlsruhe Institute of Technology (KIT), Institute of Meteorology and Climate Research (IMK-IFU), Garmisch Partenkirchen, Germany
[16]Finnish Meteorological Institute, Sodankylä, Finland
[17]National Institute of Meteorological Research, Seoul, Republic of Korea

*Correspondence to:* Lianghai Wu (l.wu@sron.nl)

**Abstract.**

In this study we present the retrieval of the column averaged dry air mole fraction of carbon dioxide ($X_{CO_2}$) from the Orbiting Carbon Observatory-2 (OCO-2) satellite observations using the RemoTeC algorithm, previously successfully applied to retrieval of greenhouse gas concentration from the Greenhouse Gases Observing Satellite (GOSAT). The $X_{CO_2}$ product has been validated with collocated ground based measurements from the Total Carbon Column Observing Network (TCCON) for almost 2 years of OCO-2 data from September 2014 to July 2016. We found that fitting an additive radiometric offset in all three spectral bands of OCO-2 significantly improved the retrieval. Based on a small correlation of the $X_{CO_2}$ error over land with fit residuals, we applied an a posteriori bias correction to our OCO-2 retrievals. In daily averaged results, $X_{CO_2}$ retrievals have a standard deviation $\sim 1.30$ ppm and a station-to-station variability of $\sim 0.40$ ppm among collocated TCCON sites. The





seasonal relative accuracy (SRA) has a value of 0.52 ppm. The validation shows relatively larger difference with TCCON over high latitude areas and some specific regions like Japan.

## 1   Introduction

Carbon dioxide ($CO_2$) concentration is rapidly increasing in the atmosphere due to fossil fuel combustion and deforesta-
tion (Prentice et al., 2001). This can lead to significant changes in climate (Cox et al., 2000; Caldeira and Wickett, 2003). Any mitigation strategy to reduce $CO_2$ in the atmosphere requires a better understanding of the global carbon cycle, especially, identifying carbon dioxide emissions from both natural and anthropogenic sources and sinks that absorb carbon dioxide. Our ability to quantify sources and sinks of $CO_2$ is still insufficient due to the sparseness of current ground-based stations (Gurney et al., 2002; Patra et al., 2003; Houweling et al., 2004; Bösch et al., 2006; Baker et al., 2010). To get a better understand-
ing of the spatial and temporal pattern of sources and sinks of $CO_2$, efforts have been made to retrieve $X_{CO_2}$ from satellite observations. The thermal infrared observations of $CO_2$ from instruments like the Atmospheric Infrared Sounder (AIRS), the Tropospheric Emission Spectrometer (TES) and the Infrared Atmospheric Sounding Interferometer (IASI) can provide $CO_2$ measurements at altitudes between 5 and 15 km (Chédin et al., 2002; Engelen et al., 2004; Crevoisier et al., 2009). These measurements have a limited sensitivity to $CO_2$ in the lower troposphere where $CO_2$ sources and sinks are located. Satellite
observations measuring in the short-wave infrared (SWIR) spectral range, however, are sensitive to $CO_2$ down to the Earth's surface in the absence of clouds and so this spectral range is used to measure $X_{CO_2}$ by several space missions. The SCanning Imaging Absorption spectroMeter for Atmospheric CHartographY (SCIAMACHY), operational between 2003 and 2012, is the pioneering instrument measuring $X_{CO_2}$ from the SWIR spectra with sensitivity in the boundary layer (Buchwitz et al., 2005). Reuter et al. (2011) showed that accurate $X_{CO_2}$ can be inferred from SCIAMACHY observations, taking atmospheric
scattering processes into account in the retrieval. The Greenhouse Gases Observing Satellite (GOSAT), in orbit since January 2009, is the first satellite primarily dedicated to monitor global atmospheric levels of $CO_2$ and $CH_4$ from space (Yokota et al., 2009). The $X_{CO_2}$ derived from GOSAT has an accuracy well below 1.0% (Butz et al., 2011; Guerlet et al., 2013b; Kulawik et al., 2017). $X_{CO_2}$ retrievals with this level of accuracy can provide valuable information on the behavior of sources and sinks of $CO_2$ (Rayner and O'Brien, 2001; Houweling et al., 2004; Guerlet et al., 2013a; Basu et al., 2014; Detmers et al., 2015). In
July 2014, NASA's Orbiting Carbon Observatory-2 (OCO-2) satellite was successfully launched. OCO-2 is designed with three standard observational modes (nadir, glint and target) for accurate monitoring of the geographic distribution of carbon dioxide sources and sinks on a regional scale (Crisp et al., 2004). By taking advantage of the target mode where many observations are acquired over ground-based validation sites, the biases in the $X_{CO_2}$ retrievals from OCO-2 measurements can be accurately evaluated. Furthermore, with a spatial sampling size of about 3 km$^2$, the number of cloud-free $X_{CO_2}$ OCO-2 observations
exceeds significantly those of previous missions.

One of the main challenges of $X_{CO_2}$ retrieval from SWIR satellite measurements is to characterize the light path through the atmosphere affected by atmospheric scattering and surface reflection (Aben et al., 2007). For this purpose, current missions include measurements in the near infrared (NIR) spectral range covering the $O_2$ A absorption band. Measurements in the



NIR and SWIR spectral bands allow for the simultaneous retrieval of carbon dioxide concentration with proper accounting of scattering properties introduced by aerosol or cirrus. Several algorithms have been developed to retrieve $CO_2$ from NIR/SWIR measurements from space, including the differential optical absorption spectroscopy (DOAS) retrieval method developed for the retrieval of SCIAMACHY (Buchwitz et al., 2000; Hönninger et al., 2004; Reuter et al., 2010), the algorithm developed

at the National Institute for Environment Studies (NIES) for GOSAT observations (Yoshida et al., 2011), the Atmospheric $CO_2$ Observations from Space (ACOS) retrieval algorithm developed for the OCO instrument and later applied to the GOSAT and OCO-2 observations (O'Dell et al., 2012; Crisp et al., 2012), the algorithm developed in the University of Leicester (UoL) (Boesch et al., 2011; Jung et al., 2016), and the RemoTeC algorithm developed by SRON Netherlands Institute for Space Research and Deutsches Zentrum für Luft- und Raumfahrt e.V. (DLR) (Hasekamp and Butz, 2008; Butz et al., 2011;

Guerlet et al., 2013b).

The operational $X_{CO_2}$ data product of the OCO-2 mission is derived with the ACOS algorithm and validated against ground-based measurements (Wunch et al., 2017) and a dataset is avaliable for assessing regional-scale sources and sinks (Eldering et al., 2017). To enhance the reliability and confidence of the data product, however, analyzing the data with independent algorithms is essential. For example, in the greenhouse gas project of ESA's Climate Change Initiative (GHG-CCI) extensive

comparisons were made between different $X_{CO_2}$ retrieval algorithms which showed similar results when comparing with TC-CON data. However, in other regions the differences were sometimes significantly larger (Dils et al., 2014). In this paper, we adapt and apply the RemoTeC retrieval algorithm, previously applied to the GOSAT measurements, to OCO-2 measurements obtained under nadir, glint and target modes and evaluate the $X_{CO_2}$ retrieval data quality with collocated ground based measurements from the Total Carbon Column Observing Network (TCCON) (Wunch et al., 2011a). To screen out too chal-

lenging soundings (i.e. clouds, high aerosol loadings, large spectral uncertainties) we optimized the a posteriori data filtering and developed an $X_{CO_2}$ bias correction based on spectral fit residuals.

The paper is organized as follows: Section 2 describes the OCO-2 data and Section 3 introduces the RemoTeC full physics retrieval algorithm including cloud screening and adjustments specific to OCO-2 type of measurements. In Section 4, we evaluate our retrieval results using collocated TCCON measurements. Here, the effect of bias correction is also discussed.

To further evaluate the RemoTeC/OCO-2 retrievals, section 5 discusses the TCCON validation of $X_{CO_2}$ data product from ACOS/OCO-2 and RemoTeC/GOSAT retrievals. Finally, Section 6 concludes the paper.

## 2   Data

The OCO-2 satellite provides measurements of sunlight backscattered by the Earth's surface and atmosphere in three channels including the molecular oxygen ($O_2$) A-band (around 0.765 $\mu$m, NIR), a weak $CO_2$ band (around 1.61 $\mu$m, SWIR-1) and a

strong $CO_2$ band (around 2.06 $\mu$m, SWIR-2) with a spectral resolution of $\sim 0.042$, $\sim 0.076$ and $\sim 0.097$ nm for the three bands, respectively, defined as the full-width at half maximum (FWHM) of the instrument spectral response. Each FWHM is oversampled by a factor 2 to 3 in the direction of dispersion. In each band, a linear polarizer is mounted in front of the imaging spectrometer and selects polarization vector parallel to the entrance slit. During operation, OCO-2 can collect observations





with high signal-to-noise ratios under nadir, glint and target modes and each sounding provides measurements in 8 footprints adjacent to each other. The typical size of one footprint is around $1.3$ km $\times$ $2.25$ km under the nadir observation mode and can be a bit larger (around $3\,\mathrm{km}^2$) for the other modes (Crisp et al., 2017).

In this study, we use version 7 OCO-2 data for the period September, 2014 to July, 2016. These data include observations
obtained under nadir, glint and target observation modes. A few percent of the pixels of the OCO-2 detectors show performance anomalies (Crisp et al., 2017) and so we exclude the corresponding spectral samplings using the mask information provided in the L1b files. Finally, only spectra are processed where at least half of the spectral samplings passes this quality check.

For validation purpose, we focus on satellite observations that are collocated with measurements from TCCON, which is a global network of ground-based instruments that can measure $X_{CO_2}$ in the atmosphere (Wunch et al., 2011a). The $X_{CO_2}$
measured by TCCON has an uncertainty better than $0.25\%$ ($\sim 1$ ppm) (Wunch et al., 2015). More information on TCCON sites including locations and operational status can be found at https://tccon-wiki.caltech.edu/. The collocation criteria between OCO-2 measurements and TCCON measurements include a geographical distance less than $5$ degrees in both latitude and longitude and a time difference less than 2 hours. Due to the high spatial sampling of OCO-2 ($24$ spectra per second over the swath), there are generally more than $150$ cloud-screened spectra available for each collocated TCCON measurement. In this
case, we use a maximum of $150$ nadir or glint spectra, which are spatially closest to TCCON site, while for target observations we select those obtained with a viewing zenith angle smaller than $30°$.

In addition to the OCO-2 spectra, the retrieval algorithm requires information on vertical profiles of pressure, temperature, humidity and surface wind speed, which are interpolated from the ECMWF (European Centre for Medium-Range Weather Forecasts) high resolution 10-day forecast analysis data on a $0.125° \times 0.125°$ latitude $\times$ longitude grid. The surface elevation
information of the OCO-2 footprint is extracted from the $90$ m digital elevation data of NASA's Shuttle Radar Topography Mission (SRTM) (Farr et al., 2007). We extrapolate the lowest ECMWF pressure point to the surface elevation provided by the SRTM data using the barometric law. To provide the algorithm initial guess of the $CO_2$ vertical concentration profiles and the $CH_4$ total column at each location, we use data from CarbonTracker and TM5 model for the year 2013 and 2010 (Peters et al., 2007; Houweling et al., 2014), respectively. The high-resolution solar irradiance data by Dr. R. Kurucz (http://kurucz.harvard.
edu/sun/irradiance2008/) is used as reference solar spectrum in the forward radiative transfer model.

## 3  Method

The RemoTeC algorithm has been described in detail by Hasekamp and Butz (2008); Butz et al. (2009, 2010) and has been applied for $CO_2$ and $CH_4$ retrievals from GOSAT measurements (Butz et al., 2011; Schepers et al., 2012; Guerlet et al., 2013b). In the following, we assume that the OCO-2 radiance measurements $\boldsymbol{y}$, comprising of measurements in all three bands, can be
described by a forward radiative transfer model $\boldsymbol{F}$ via,

$$\boldsymbol{y} = \boldsymbol{F}(\boldsymbol{x}, \boldsymbol{b}) + \boldsymbol{e} \tag{1}$$





Here, $\boldsymbol{x}$ is the state vector containing all parameters to be retrieved and $\boldsymbol{b}$ includes a set of auxiliary input parameters. The error term $\boldsymbol{e}$ contains uncertainties in both instrument and forward model. To infer $X_{CO_2}$, RemoTeC resolves Eq.1 with respect to the state vector $\boldsymbol{x}$.

The OCO-2 instrument measures the backscattered sunlight in a single polarization direction, and so the forward model for spectral sampling $i$ reads,

$$F_i(\boldsymbol{x}, \boldsymbol{b}) = m_{11}I_i + m_{12}Q_i + m_{13}U_i \tag{2}$$

where $I_i$, $Q_i$ and $U_i$ are the first three stokes parameters of a line-by-line top of the model atmosphere spectrum convolved with the OCO-2 instrument spectral response function. The elements of the Muller matrix $m_{11}$, $m_{12}$, and $m_{13}$ describe the instrument polarization sensitivity depending on the illumination and observing geometries of the OCO-2 instrument. For the simulation of the line-by-line spectra, we employ the LINTRAN vector radiative transfer model Hasekamp and Landgraf (2002, 2005a); Schepers et al. (2014). To simulate efficiently the spectral dependence of the Stokes parameter $I$, $Q$ and $U$, defined at the top of the model atmosphere, the multiple scattering calculations are performed following the k-binning approach of Hasekamp and Butz (2008) while the single scattering is calculated line-by-line. In the algorithm, the model atmosphere is divided into 36 sub-layers for the radiative transfer calculation and further divided into 72 sub-layers for absorption cross-section calculation which is highly dependent on temperature and pressure.

Since the measurement $\boldsymbol{y}$ does not contain sufficient information to retrieve all elements of state vector $\boldsymbol{x}$, the algorithm employs a Phillips-Tikhonov regularization scheme to solve the minimization problem iteratively (Phillips, 1962; Tikhonov, 1963; Hasekamp and Landgraf, 2005b),

$$\hat{\boldsymbol{x}} = \min_{\boldsymbol{x}}(\|\boldsymbol{S}_y^{-\frac{1}{2}}(\boldsymbol{F}(\boldsymbol{x}) - \boldsymbol{y})\|^2 + \gamma\|\boldsymbol{W}(\hat{\boldsymbol{x}} - \boldsymbol{x}_a)\|^2), \tag{3}$$

where $\boldsymbol{S}_y$ is the diagonal measurement error covariance matrix that contains the measurement error estimate, $\boldsymbol{x}_a$ is a prior state vector, $\gamma$ is the regularization parameter and $\boldsymbol{W}$ is the weighting matrix making the side constraint dimensionless. The value for $\gamma$ is fixed such that the degree of freedom for signal (DFS) for the carbon dioxide profile is in the range 1.0-1.5. To avoid diverging retrievals, following a Gauss-Newton scheme (Rodgers, 2000) a filter factor ($\Lambda = \frac{1}{1+\xi}, \xi \geq 0$) is also introduced to limit the update of the state vector per iteration step. More details on this aspect of the RemoTeC implementation can be found in Butz et al. (2012). The retrieval is considered successful when following conditions are all met: (1) the update of the state vector $\boldsymbol{x}$ become smaller than its theoretical uncertainty; (2) the step-size parameter $\xi$ has reached 0; (3) the state vector elements have never reached unrealistic values during the iteration.

The forward model assumes the land surface reflection to be Lambertain, whereas ocean surface reflection is modeled using a wind speed dependent Cox-and-Munk reflection model (Cox and Munk, 1954) with a wavelength dependent Lambertian term added to it. Oxygen absorption lines in the A band are calculated by a spectroscopic model that accounts for line mixing and collision-induced absorption processes (Tran and Hartmann, 2008). Absorption lines of $CO_2$ are modelled accordingly to the HITRAN 2008 spectroscopic data base, by taking line-mixing into account (Rothman et al., 2009; Lamouroux et al., 2010). HITRAN 2008 is also used to model absorption lines of $CH_4$ and $H_2O$ assuming a Voigt lineshape model. In the retrieval, we





treat aerosol as spherical particles with a constant refractive index $(1.400 - 0.003i)$ over the whole OCO-2 spectral range. The aerosol size distribution is described by a power law function $(n(r) \propto r^{-\alpha_s})$ with size parameter $\alpha_s$ while the aerosol height profile is assumed to be Gaussian with a central height parameter $z_s$ and a fixed geometric width of 2 km. Based on this aerosol model, we calculate the optical properties of aerosol particles using the tabulated kernels of Dubovik et al. (2006).

In the retrieval, the state vector $x$ includes the 12-layer profile of $CO_2$ sub-column number densities along with total column number densities of interfering absorbers $CH_4$ and $H_2O$, surface parameters including a second order spectral dependence of the Lambertian surface albedo in all OCO-2 bands. Moreover, $x$ contains the aerosol size parameter $\alpha_s$ of the power-law distribution, the total column density of aerosol particles, and the central height parameter $z_s$ of the Gaussian height distribution. Finally, in all three bands we infer an intensity offset, a first order spectral shift of the Earth radiance spectrum and a spectral

shift of the solar reference spectrum. To initialize the retrieval, we choose an aerosol total column, which corresponds to an aerosol optical depths of 0.1 in the NIR spectral band, a size parameter $\alpha_s = 4.5$ and an aerosol layer height $z_s = 3000$ m. After convergence, the spectral fit residuals are generally less than 1.0% with a typical chi squared of 3.0.

Since clouds are not considered in RemoTeC, a cloud screening of the OCO-2 data is required before performing full physical retrieval. For this purpose, we apply the screening approach similar to Taylor et al. (2016) in our algorithm and compare

columns of $O_2$, $CO_2$ and $H_2O$, which are retrieved independently for a non-scattering atmosphere from the NIR, SWIR-1, and SWIR-2 bands of OCO-2, respectively. When neglecting cloud and aerosol scattering a large deviation can be introduced between $CO_2$ and $H_2O$ columns retrieved from SWIR-1 and SWIR-2 bands due to different light path sensitivity. Similarly, for scenes with larger photon path-length modification, the retrieved $O_2$ column will deviate more from the $O_2$ column provided by the ECMWF. Here, around 30% of total soundings are identified as cloud-free cases by the cloud screening. Apart from cloud

screening, observations with solar zenith angle $> 70°$ and large surface roughness (standard deviation of surface elevation $> 75$ m) are also excluded before performing the operational retrievals.

## 4  Validation with TCCON

In this section, we evaluate the $X_{CO_2}$ retrieved from OCO-2 measurements using the RemoTeC algorithm against ground-based measurements at a comprehensive set of TCCON stations. Figure 1 shows an example of validation between RemoTeC/OCO-2

retrievals and TCCON measurements over Lamont and Darwin stations. To evaluate our retrieval quality, we use the bias ($b$) as the mean difference between collocated TCCON and OCO-2 retrievals, the sounding precision ($\sigma$) as the standard deviation of the difference and the station-to-station variability ($\sigma_s$) as the standard deviation of the biases for different TCCON stations. Here, retrievals over land and ocean are evaluated separately. Land retrievals include observations obtained under nadir and glint modes and ocean retrievals only include observations under glint mode. Target mode observations, mostly performed

coincidentally around TCCON sites over land, are evaluated separately. Moreover, the standard deviation over all seasonal bias results, known as "seasonal relative accuracy"(SRA) introduced by Dils et al. (2014), are also derived for all three types of retrievals. The SRA value is a good indicator of the variability of the bias in both space and time. In the following validation, we assume that TCCON measurements themselves are consistent over all stations with a station-to-station variability of zero.





However, as discussed by Kulawik et al. (2016); Buchwitz et al. (2017b), individual stations have a year-to-year variability of $\sim 0.3$ ppm and the overall TCCON $X_{CO_2}$ uncertainty is around $0.4$ ppm (1-sigma). Although some limitations may exist, TCCON measurements are the most appropriate validation data product for satellite observations. Here, we exclude stations located either close to source region such as Caltech or on very high latitude such as Eureka. Land retrievals obtained over Reunion Island, located within areas with significant topography and an active volcano, will also not be used for validation.

## 4.1 Filters and bias correction

We first compare our retrieval results with collocated TCCON data to establish a set of values for the filters shown in Table 1 to screen out retrievals with larger uncertainties. In our retrieval, around 83%, 81% and 72% of cloud-free cases successfully converge and, after applying the filters in Table 1, 66%, 50% and 47% of retrievals remain with good quality in cloud-screened target, land and ocean types of measurements, respectively.

Similar to the work of Butz et al. (2011); Guerlet et al. (2013b), we apply filters to reject retrievals with bad quality of fit ($\chi^2 > 7.0$, $\chi^2_{1st} > 7.0$ or not converged with number of iterations $> 30$), or with high aerosol loading ($\tau_{0.765} > 0.35$), or with extreme aerosol parameters ($\alpha_s < 3.5$, $\alpha_s > 8.0$ or aerosol ratio parameter $> 300$ m), or with surface types like snow or ice (blended albedo $> 0.9$). Here, $\chi^2$ is defined as $1/N \sum_{i=1}^{N} (\frac{(y(i)-F(i))}{\delta_i})^2$, in which $N$ is number of measurements, $y(i)$ is the OCO-2 measurement, $F(i)$ is the simulated result and $\delta_i$ is the uncertainty of the OCO-2 measurements. In OCO-2 retrievals, intensity offset parameters are fitted for all the three spectral windows and we use the ratio between retrieved intensity offset and mean spectral radiance to filter out soundings with larger spectral uncertainties. Here, target retrievals have the same filter settings as land retrievals.

Ocean glint measurements require different filter settings because of their different sensitivity due to unique viewing geometry and different surface properties. Moreover, in the measured radiance of ocean glint measurements, the contribution from aerosol scattering is negligible when compared with that from ocean surface reflection. As a consequence, the measured radiances are mainly sensitive to ocean reflection and aerosol layer extinction properties. Aerosol filter settings used here are different from land retrievals due to the limitation of aerosol information and aerosol parameters like particle size and layer height usually retain their prior values.

When comparing our retrieval results with collocated TCCON measurements, we look for possible correlations of errors with instrumental, geophysical, meteorological and retrieved parameters. In this paper, a positive bias means $X_{CO_2}$ is overestimated by the RemoTeC/OCO-2 retrievals. Figure 2 shows that only a small overall bias of $0.31$, $0.37$, and $0.70$ ppm exist in the RemoTeC/OCO-2 retrievals for target, land and ocean types of retrievals, respectively. However, if we look at retrievals from 8 individual footprints within a swath separately, the $X_{CO_2}$ retrievals show statistically significant differences on overall biases ranging from $-0.25$ to $0.65$ ppm with a standard deviation of $0.3$ ppm. These biases arise from uncertainties in the L1 processing depending on the viewing direction in across flight direction and have to be removed before performing an overall bias correction. To identify the footprint dependent biases, we use target mode observations when all 8 footprints in one sounding frame converged, which provides around 7000 available retrievals per footprint. By using a large amount of target observations





we can reduce the uncertainties in the footprint-to-footprint bias estimation. Here, we assume a constant $X_{CO_2}$ field in across track direction. The estimated swath-dependent biases, as shown in Fig 3, are directly subtracted from each footprint.

After removing the swath-dependent biases, a bias dependence on the fit residual $\chi^2$ in SWIR-1 band is found for RemoTeC/OCO-2 retrievals over land. As shown in Fig. 4, a typical $\chi^2$ in SWIR-1 band is around 2.0 and the correlation cofficient is 0.20. This

bias can be attributed to many factors like spectroscopic errors, inconsistent aerosol assumptions and instrument or algorithm uncertainties. We correct this bias by

$$XCO_2^{corr} = XCO_2(d + k \cdot \chi^2), \tag{4}$$

where the coefficients $k = -0.001261$ and $d = 1.001938$ are derived with a linear regression fit through the difference between individual retrievals and TCCON measurements. This correction reduces the error correlation with most parameters in Table 1

such as surface albedo in the NIR band(albedo_NIR), solar zenith angle and degrees of freedom for signal, even though these parameters are not used in the bias correction and the remaining correlations with related parameters are generally less or around 0.10. After applying this bias correction the swath-dependent biases remains low around $-0.1$ ppm with a standard deviation of 0.01 ppm.

For ocean glint retrievals, we only subtract the swath-dependent bias and a constant bias of 0.65 ppm from the $X_{CO_2}$ results.

The constant bias is obtained by validating retrieval results with collocated TCCON measurements from sites as listed in Fig. 7. The $X_{CO_2}$ swath-dependent bias for ocean glint observations is very similar to the one of $X_{CO_2}$ target observations and so the same correction is applied.

Overall, with the bias correction in Eq. 4 the sounding precisions $\sigma$ are slightly improved by $\sim 0.1$ ppm for land retrievals in Fig. 2. The land and ocean bias corrections are developed for reducing globally-relevant biases and thus geographically related

or time dependent biases may remain in the results and need further investigation.

### 4.2   TCCON validation

For a detailed validation of the bias corrected $X_{CO_2}$ product, we will evaluate the $X_{CO_2}$ retrieved from OCO-2 target, land and ocean measurements using the RemoTeC algorithm for different TCCON stations separately. The average of the retrieved $X_{CO_2}$ is compared with the corresponding TCCON average values. We exclude cases where less than 5 individual data points are available within 2 hours in either OCO-2 retrievals or TCCON data. To evaluate the retrieval quality, we take into account

the bias ($b_a$), standard deviation ($\sigma_a$), station-to-station variability ($\sigma_s$) and seasonal relative accuracy (SRA) against TCCON measurements station by station. Here, the station-to-station variability is an important evaluation parameter known as a measure of regional-scale accuracy, which is most important for estimating $CO_2$ surface-to-atmosphere fluxes on regional scales. The SRA value further indicates the potential bias variation in both space and time. Moreover, we study the effect of the bias

correction by analysing the retrieval performance station-by-station.

Figures 5, 6 and 7 show the overall comparisons between RemoTeC/OCO-2 retrievals after bias correction and TCCON measurements for target, land and ocean retrievals, respectively. In the daily averaged results, the bias and standard deviation ($b_a$, $\sigma_a$) are ($-0.07$, 1.24), (0.00, 1.36), and (0.00, 1.20) ppm for target, land and ocean retrievals, respectively. Before bias





correction, the mean biases are $0.51$, $0.44$ and $0.62$ ppm for the above three type of retrievals, respectively. The bias correction mainly improves the mean bias though the standard deviations are also reduced by $\sim 0.1$ ppm for land retrievals.

Figure 8, 9 and 10 show the bias ($b_a$) at each TCCON site as a function of its latitude for the target, land and ocean types of retrievals. The mean ($\overline{b_a}$) and the standard deviation ($\sigma_s$) of all the biases are also derived. Stations with less than 5 valid points have been excluded from the analysis. The number of stations used in the validation are 10, 17 and 18 for target, land and ocean retrievals, respectively. Within those stations, most of them have a bias less than $0.5$ ppm for both land and ocean retrievals.

In Fig. 8, the remaining $X_{CO_2}$ bias for target observations varies from $-0.81$ ppm (Tsukuba, Japan) to $0.47$ ppm (Lauder, New Zealand). The developed bias correction reduces the station-to-station variability from $0.54$ ppm to $0.35$ ppm. The effect of the bias correction is largest for Lamont, Dryden and Darwin stations ($> 0.50$ ppm on the mean station bias) while in other stations the difference is small. Land retrievals as shown in Fig 9, validated among 17 stations, have a station-to-station variability of $0.41$ ppm. The remaining bias varies from $-0.66$ ppm (Lamont, OK(USA)) to $1.03$ ppm (Sodankyla, Finland). Here, most stations have similar biases as found for the corresponding target observations. The bias correction also helps to reduce the station-to-station variability for land retrievals although not that much. Among all the stations, Tsukuba station in Japan have relatively larger standard deviation of $2.07$ ppm. For retrievals in Figs 8 and 9, there is a tendency that validations over stations in higher latitude areas have relatively larger biases in both northern and southern hemispheres. In addition, target observations have a smaller station-to-station variability than land observations although different TCCON stations are involved.

For ocean retrievals, since we only subtract swath dependent bias and a mean bias, the station-to-station variability ($0.44$ ppm) is the same before and after bias correction. The biases vary from $-0.86$ ppm (Saga, Japan) to $0.75$ ppm (Bremen, Germany). There is no clear indication of latitude dependent bias variation.

Moreover, we investigated temporal variations in RemoTeC/OCO-2 $X_{CO_2}$ retrievals. As showin in Fig.1, seasonal $X_{CO_2}$ variation features in the northern hemisphere can be well captured by both RemoTeC/OCO-2 retrievals and TCCON measurements. At the southern hemisphere, the $X_{CO_2}$ is more stable throughout the whole time range. Fig. 11 shows the time series of $X_{CO_2}$ difference between TCCON measurements and $X_{CO_2}$ retrievals from OCO-2 target, land and ocean types of measurements. At most stations, no time-dependent biases can be clearly observed. For some stations in the northern hemisphere like Sodankyla, Bremen and Paris, time-dependent features can also be attributed to inhomogeneous seasonal data distribution. There are some outliers in $X_{CO_2}$ retrievals from both land and ocean glint observations, such as those at the Tsukuba over land and Lauder over ocean, that need further investigation.

Finally, we check the "seasonal relative accuracy"(SRA) which is derived for all three types of observations. For each station, all the data regardless of the year are sorted into four intervals of a calendar year. Table 2 summarizes seasonal bias per station, standard deviation of biases per season, seasonal variability ("Seas") and the SRA value. The derived SRA of $0.52$ ppm is close to the requirement of $0.50$ ppm as discussed by Dils et al. (2014). Here, the developed bias correction helps to improve the SRA from $0.60$ to $0.52$ ppm. In stations where seasonal variability can be caculated, the value is generally around $0.30$ ppm except stations Rikubetsu ($0.71$ ppm) and Saga ($0.43$ ppm) in Japan. In Table 2 the SRA values are mainly driven by large negative





biases from Rikubetsu, Tsukuba and Saga stations in Japan. Further investigations are needed to diagnose the remaining larger biases in certain season over stations in Japan.

### 4.3 Importance of intensity offset

As mentioned in section 3, the implementation of the RemoTeC algorithm, used in this study, fits an intensity offset for all three OCO-2 bands. To identify its importance, we performed the same retrieval as in Fig. 6 but without fitting intensity offset in the SWIR-1 and SWIR-2 bands. Figure 12 shows that without fitting intensity offsets in the SWIR-1 and SWIR-2 bands the validation exhibits a negative bias of $-2.95$ ppm and the standard deviation increased by $\sim 0.5$ ppm.

As shown in Fig 13, in the SWIR-1 and SWIR-2 bands the fitted intensity offsets are propotial to the mean radiance with a slope of $0.0025$ and $0.0035$, respectively. This slope is generally two times larger than that of noise. Generally, the fitted intensity offset in these two bands are $\sim 0.4\%$ of the corresponding mean radiance. There are no time-dependent features in the fitted intensity offset. The intensity offset in the $O_2$ A band shows a less strong dependence on the signal level itself. Here, it could be a compensation to solar-induced chlorophyll fluorescence (SIF) since currently it is not fitted in the retrieval. Moreover, it could also be partly introduced by light reflection by degraded anti-reflection coating on the focal plane array (Crisp et al., 2017). However, this can not explain the amount of intensity offset retrieved in our algorithm for the SWIR-1 and SWIR-2 bands since for those channels much thicker and higher index anti-reflection coatings are used (Crisp et al., 2017). Potential causes could be straylight from reflection of nearby ground pixels or from components of the optical system.

### 5 Discussion

As we mentioned before, OCO-2 level-2 products delivered by the ACOS retrieval algorithm are also validated with collocated TCCON data by Wunch et al. (2017). Before comparing our results with the results in Wunch et al. (2017), we need to point out several differences of the validation approach by Wunch et al. (2017) and our study: (1) The considered time range of the study by Wunch et al. (2017) is from September, 2014 to January, 2017; (2) A collocation criterion of $5°$ in latitude and $10°$ in longitude is applied for most stations, but for Caltech and Dryden and those located on the southern hemisphere, a specific local collocation criterion is employed; (3) daily median values of both OCO-2 retrievals and TCCON are used for comparison; (4) observations over land under nadir and glint modes are validated separately;(5) the employed filter settings and bias corrections are also different from here.

Albeit with so many differences, we still see a lot of common aspects when looking at the standard deviation and station-to-station variability in Wunch et al. (2017). For example, for the results under warn level 11 (the best 50% of the total L2 data, see Mandrake et al. (2015) for more details on warn level) the standard deviation of the difference (OCO-2-TCCON) for land retrievals is around 1.3 ppm. Looking at station-to-station variability for ACOS land retrievals, the $\sigma_s$ is $\sim 0.45$ ppm over 12 stations. For ocean glint retrievals, the $\sigma_s$ is $0.46$ ppm over 9 stations. These values are more or less the same, albeit a bit higher, as what we see in the validation in Figs. 9 and 10.





$X_{CO_2}$ retrievals from GOSAT measurements using the RemoTeC algorithm have also been validated with TCCON data (Butz et al., 2011; Guerlet et al., 2013b; Dils et al., 2014; Buchwitz et al., 2017b). There are several improvements on the RemoTeC/GOSAT $X_{CO_2}$ retrieval quality since the first report by Butz et al. (2011). Here we will use the latest results reported by Buchwitz et al. (2017a). It should be noted that there are quite some differences between RemoTeC/GOSAT and

RemoTeC/OCO-2 including instrument polarization sensitivity, collocation criteria, filtering options and so on. In the validation between RemoTeC/GOSAT $X_{CO_2}$ retrievals and TCCON data, the sounding precision is 1.9 ppm with a station-to-station variability (estimated over 12 stations) of 0.43 ppm. The derived SRA is 0.51 ppm. Looking at those overall statistic values, there are no significant differences between $X_{CO_2}$ retrievals from RemoTeC/OCO-2 and RemoTeC/GOSAT. However, further investigation is needed to identify the difference between $X_{CO_2}$ retrievals from those two satellites, especially over regions

where TCCON data is not available.

## 6  Conclusions

In this paper, we extended and adapted the full physics retrieval algorithm RemoTeC, previously applied to GOSAT, for OCO-2 satellite measurements. The algorithm was applied to OCO-2 nadir, glint and target observations obtained over land and ocean (glint only). We defined both an a posteriori data filtering approach and bias correction as a function of the swath position by

comparing with TCCON. Additionally, we introduced a linear bias correction for land observations as a function of the spectral fit quality. Comparison of the retrieved $X_{CO_2}$ with collocated ground-based TCCON stations showed that for both land and ocean observations our retrieval results exhibit a residual bias less than 0.10 ppm with a standard deviation around 1.30 ppm (for daily means) and a station-to-station variability variation around 0.40 ppm. Among the individual TCCON stations, the biases are generally less than 0.50 ppm. In land retrievals, middle to high latitude areas have relatively larger biases and in ocean

retrievals no latitude-dependent bias can be clearly seen. The target observations have a station-to-station variability around 0.35 ppm which approaches the systematic error required for regional $CO_2$ source/sink determination (Chevallier et al., 2005; Houweling et al., 2010; Chevallier et al., 2014b, a). The better comparison with TCCON for target mode retrievals compared to regular land retrievals could be attributed to the fact that under the target mode the OCO-2 satellite is directly looking at the place where TCCON sites are located and this provides a better collocation and therefore prevents apparent biases caused by

local $X_{CO_2}$ variations. Time series validation indicates that RemoTeC/OCO-2 retrieval results can well capture the seasonal cycle of $X_{CO_2}$ in both hemispheres and no time-dependent bias can be clearly observed in the retrieval. The seasonal relative accuracy investigated over 66 time intervals of collocated stations has a value of 0.52 ppm. Most of stations have a seasonal variability around 0.30 ppm except for those in Japan. For the $X_{CO_2}$ retrieval from OCO-2 measurements, we see that intensity offsets need to be fitted for all three bands otherwise a larger bias (2.50 ppm) and standard deviation (0.50 ppm) would be

introduced in the results.





## 7 Data availability

The OCO-2 data (version 7) used here were produced by the OCO-2 project at the Jet Propulsion Laboratory, California Institute of Technology, and obtained from the OCO-2 data archive maintained at the NASA Goddard Earth Science Data and Information Services Center (https://daac.gsfc.nasa.gov/). TCCON data were obtained from the TCCON Data Archive, hosted by

5  the Carbon Dioxide Information Analysis Center (CDIAC) http://tccon.ornl.gov/ at that time. Since October 2017, the TCCON Data Archive is hosted by CaltechDATA, California Institute of Technology, CA (US), doi:10.14291/tccon.archive/1348407. The RemoTeC/OCO-2 $X_{CO_2}$ retrievals used in this paper are available upon request from Lianghai Wu (l.wu@sron.nl).

*Acknowledgements.* This research was funded by the Netherlands Space Office as part of the User Support Programme Space Research under project ALW-GO/15-23. We are grateful to Université Pierre et Marie Curie, French space agency CNES and Région Île-de-France for

10  their financial contributions as well as to Institut Pierre-Simon Laplace for support and facilities.



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





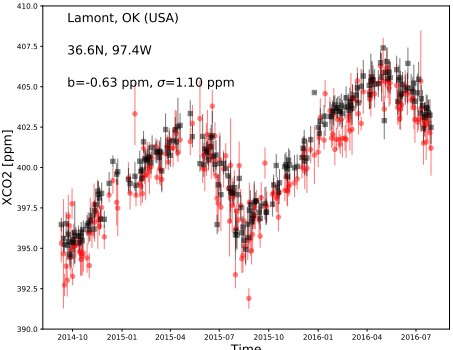 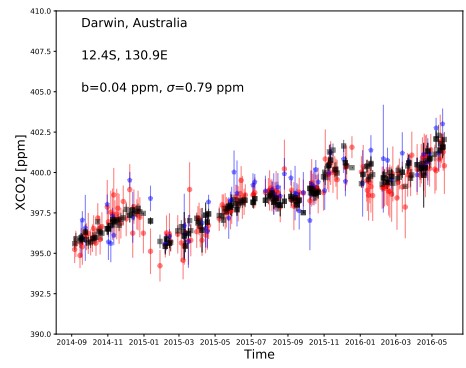

**Figure 1.** Time variation of $X_{CO_2}$ retrievals from OCO-2 observations over land (red dots) and ocean (blue pentagon) and collocated TCCON measurements (black square) for Lamont and Darwin stations. Standard deviation of individual TCCON measurement and satellite retrievals are presented with the length of bar. In each subplot, the mean bias ($b$) and standard deviation($\sigma$) of the difference between RemoTeC/OCO-2 retrievals and TCCON measurements and site location in latitude and longitude are included. The shown results here are bias-corrected data.

Wennberg, P. O., Wunch, D., Roehl, C., Blavier, J.-F., Toon, G. C., and Allen, N.: TCCON data from Lamont (US), Release GGG2014.R1, TCCON Data Archive, hosted by CaltechDATA, https://doi.org/10.14291/tccon.ggg2014.lamont01.R1/1255070, 2016.

Wunch, D., Toon, G. C., Blavier, J.-F. L., Washenfelder, R. A., Notholt, J., Connor, B. J., Griffith, D. W., Sherlock, V., and Wennberg, P. O.: The total carbon column observing network, Philosophical Transactions of the Royal Society of London A: Mathematical, Physical and

Engineering Sciences, 369, 2087–2112, 2011a.

Wunch, D., Wennberg, P., Toon, G., Connor, B., Fisher, B., Osterman, G., Frankenberg, C., Mandrake, L., O'Dell, C., Ahonen, P., et al.: A method for evaluating bias in global measurements of CO 2 total columns from space, Atmospheric Chemistry and Physics, 11, 12 317–12 337, 2011b.

Wunch, D., Toon, G. C., Sherlock, V., Deutscher, N. M., Liu, C., Feist, D. G., and Wennberg, P. O.: The Total Carbon Column Observing

Network's GGG2014 Data Version, Carbon Dioxide Information Analysis Center, Oak Ridge National Laboratory, Oak Ridge, Tennessee, USA, available at: doi, 10, https://doi.org/10.14291/tccon.ggg2014.documentation.R0/1221662, 2015.

Wunch, D., Wennberg, P. O., Osterman, G., Fisher, B., Naylor, B., Roehl, C. M., O'Dell, C., Mandrake, L., Viatte, C., Kiel, M., Griffith, D. W. T., Deutscher, N. M., Velazco, V. A., Notholt, J., Warneke, T., Petri, C., De Maziere, M., Sha, M. K., Sussmann, R., Rettinger, M., Pollard, D., Robinson, J., Morino, I., Uchino, O., Hase, F., Blumenstock, T., Feist, D. G., Arnold, S. G., Strong, K., Mendonca, J., Kivi,

R., Heikkinen, P., Iraci, L., Podolske, J., Hillyard, P. W., Kawakami, S., Dubey, M. K., Parker, H. A., Sepulveda, E., García, O. E., Te, Y., Jeseck, P., Gunson, M. R., Crisp, D., and Eldering, A.: Comparisons of the Orbiting Carbon Observatory-2 (OCO-2) $X_{CO_2}$ measurements with TCCON, Atmospheric Measurement Techniques, 10, 2209–2238, https://doi.org/10.5194/amt-10-2209-2017, 2017.

Yokota, T., Yoshida, Y., Eguchi, N., Ota, Y., Tanaka, T., Watanabe, H., and Maksyutov, S.: Global concentrations of CO2 and CH4 retrieved from GOSAT: First preliminary results, Sola, 5, 160–163, 2009.

Yoshida, Y., Ota, Y., Eguchi, N., Kikuchi, N., Nobuta, K., Tran, H., Morino, I., and Yokota, T.: Retrieval algorithm for $CO_2$ and $CH_4$ column abundances from short-wavelength infrared spectral observations by the Greenhouse gases observing satellite, Atmospheric Measurement Techniques, 4, 717–734, https://doi.org/10.5194/amt-4-717-2011, http://www.atmos-meas-tech.net/4/717/2011/, 2011.





**Table 1.** Settings of the filters used for excluding RemoTeC/OCO-2 $X_{CO_2}$ retrievals. The sign '-' indicates using the same option as in land retrievals.

| Parameter | Definition | Allowed Range | |
| --- | --- | --- | --- |
| | | Land | Ocean |
| sza | Solar zenith angle | val$\leq 70°$ | - |
| vza | Viewing zenith angle | val$\leq 45°$ | - |
| iter | Number of retrieval iterations | val$\leq 30$ | - |
| dfs | Degrees of Freedom for Signal for $CO_2$ | val$\geq 1.0$ | - |
| $\chi^2$ | Overall fit residuals | val$\leq 7.0$ | - |
| $\chi^2_{1st}$ | Fit residuals in $O_2$ A-band | val$\leq 7.0$ | - |
| Blended albedo* | $2.4\times$albedo_NIR - $1.13\times$albedo_SWIR-2 | val$\leq 0.9$ | None |
| alb$_2$ | Added Lambertian term in SWIR-2 band | None | val$\leq 0.065$ |
| sev | Surface elevation variation | val$\leq 75$ m | None |
| $\alpha_s$ | Aerosol size parameter | $3.5 \leq$val$\leq 8.0$ | $3.5 \leq$val$\leq 5.5$ |
| $\tau_{0.765}$ | Aerosol optical depth in $O_2$ A-band | val$\leq 0.35$ | val$\leq 0.55$ |
| Aerosol ratio parameter | $\tau_{0.765}*z_s/\alpha_s$, $z_s$ is aerosol layer height | val$\leq 300$ m | - |
| Xerr | Retrieval uncertainty for $X_{CO_2}$ | val$\leq 2.0$ ppm | - |
| Ioff$_1$ | Fitted Intensity offset ratio in NIR band | $-0.005 \leq$val$\leq 0.015$ | - |
| Ioff$_2$ | Fitted Intensity offset ratio in SWIR-1 band | $-0.001 \leq$val$\leq 0.015$ | - |
| Ioff$_3$ | Fitted Intensity offset ratio in SWIR-2 band | $-0.001 \leq$val$\leq 0.015$ | - |

*The blended albedo filter was first introduced in Wunch et al. (2011b).



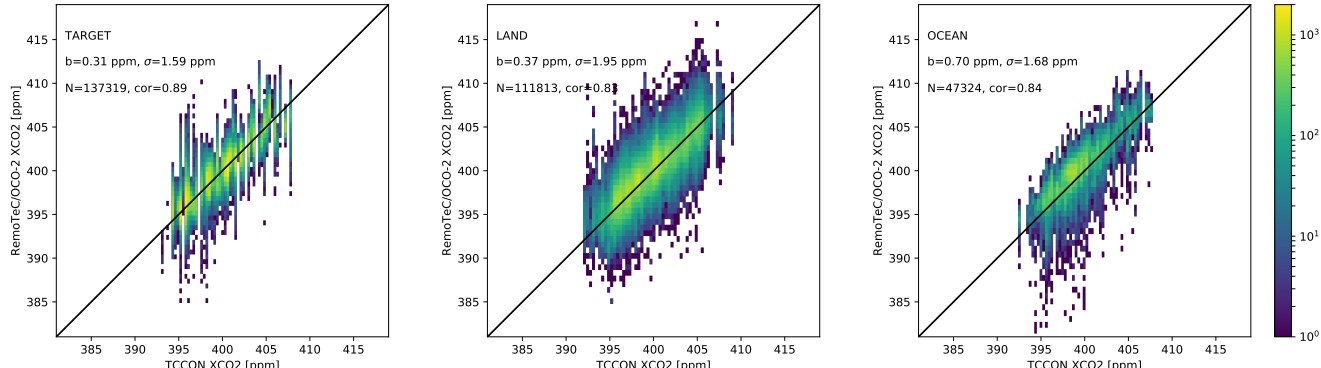

**Figure 2.** Validation of individual $X_{CO_2}$ retrieved from OCO-2 measurements with collocated TCCON data before bias correction. Here, target retrievals are separated intentionally from land retrievals results which thus only include measurements obtained under nadir and glint modes. Ocean retrievals only include glint mode observations over ocean. For retrievals collocated with multiple TCCON stations, we use data from the closest station. The bias (b), sounding precision ($\sigma$), number of points (N), the Pearson correlation coefficient (cor) and one-to-one line are included. Different colors represent the frequency of point occurrence.

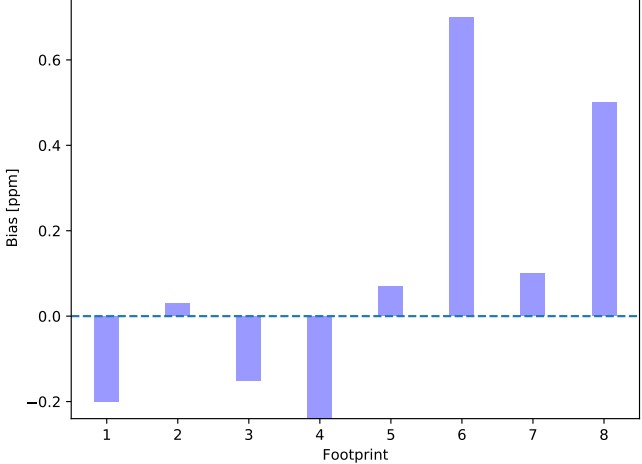

**Figure 3.** Estimated swath-dependent biases using Target mode observations.





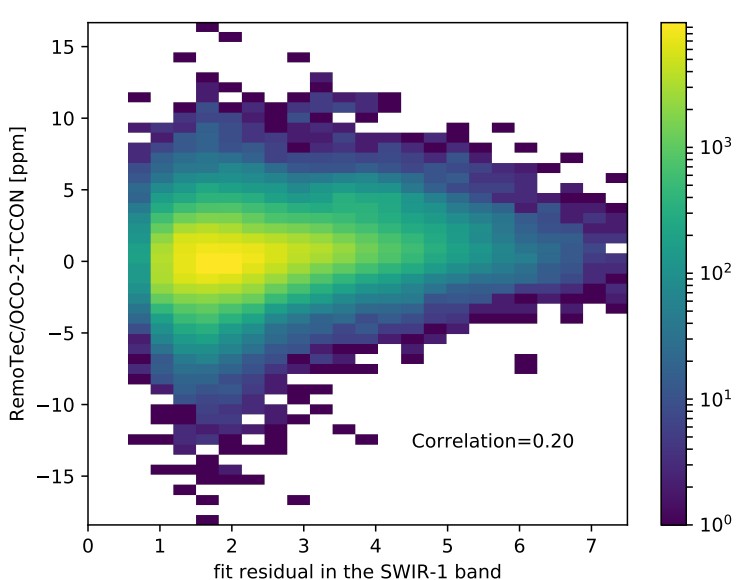

**Figure 4.** Error on $X_{CO_2}$ retrievals as a function of the fit resudual in the SWIR-1 band. Different colors represent the frequency of point occurrence.





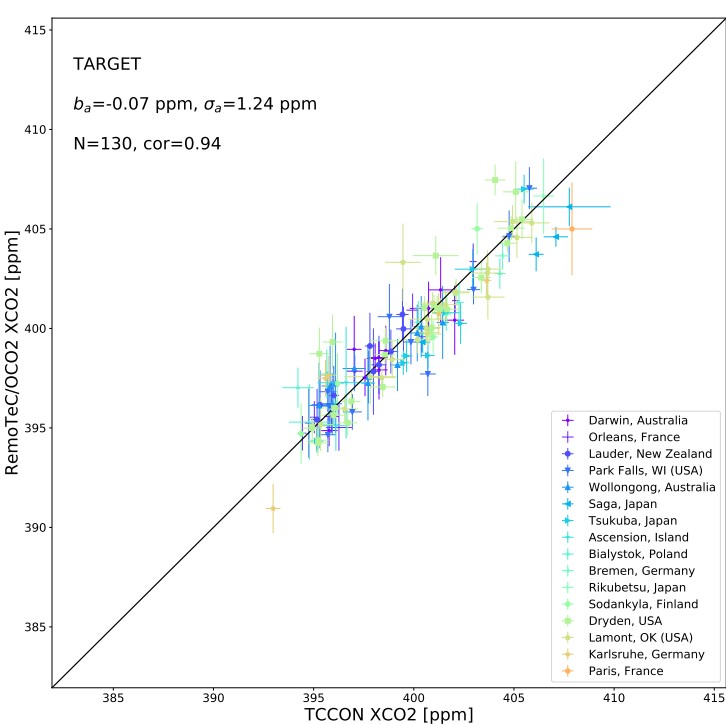

**Figure 5.** Validation of averaged $X_{CO_2}$ retrieved from OCO-2 target measurements with collocated TCCON data. The retrieval results shown here are averages of single soundings per station within 2 hours. The standard deviation of individual TCCON data and that of RemoTeC/OCO-2 retrievals are presented with error bars. The bias ($b_a$), standard deviation ($\sigma_a$), number of points ($N$), the Pearson correlation coefficient (cor) and one-to-one line are included.



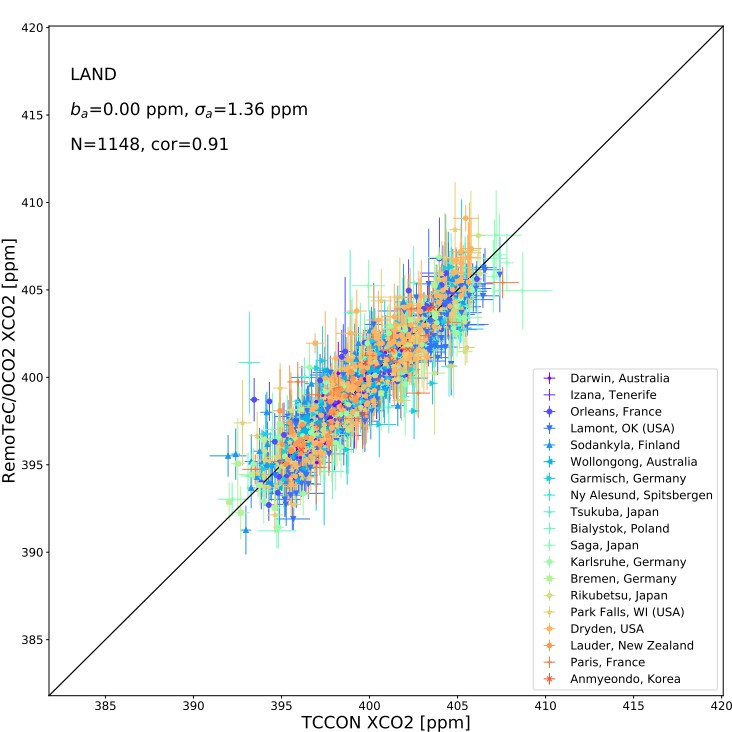

**Figure 6.** Same as Fig. 5, but for OCO-2 land type measurements obtained under nadir and glint modes.





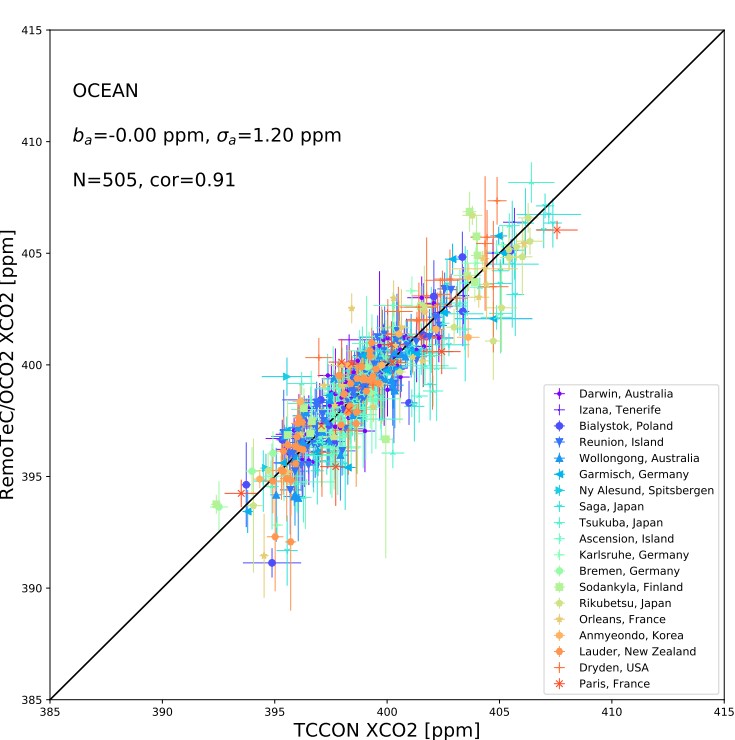

**Figure 7.** Same as Fig. 5, but for OCO-2 ocean type measurements obtained under glint mode.




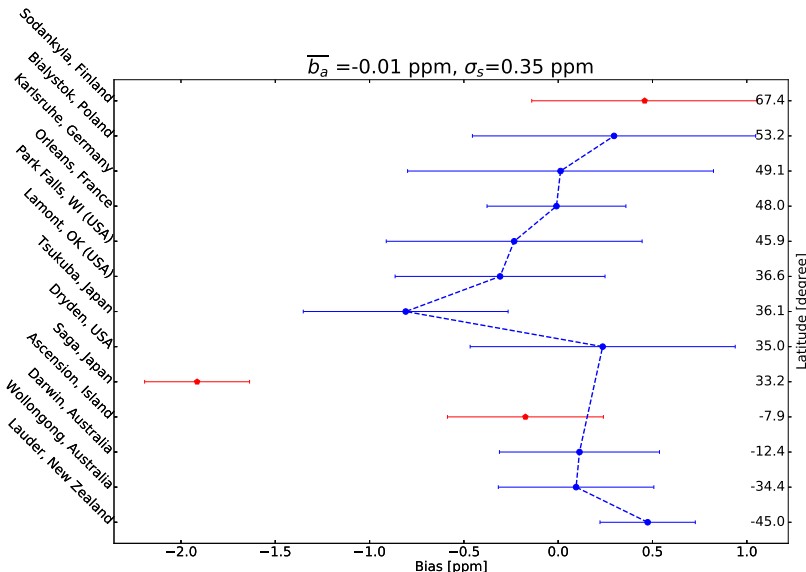

**Figure 8.** The dependence of the bias between RemoTeC/OCO-2 target $X_{CO_2}$ retrievals coincident with TCCON data on the latitude of each station. Shown are the averaged results for bias-corrected $X_{CO_2}$ retrievals. Stations with less than 5 collocation points (marked with red pentagon) should be interpreted with care and are therefore excluded from the calculation of the derived parameters including mean bias ($\overline{b_a}$) and the station-to-station variability ($\sigma_s$). The length of each bar represents the standard deviation of the difference at each station.

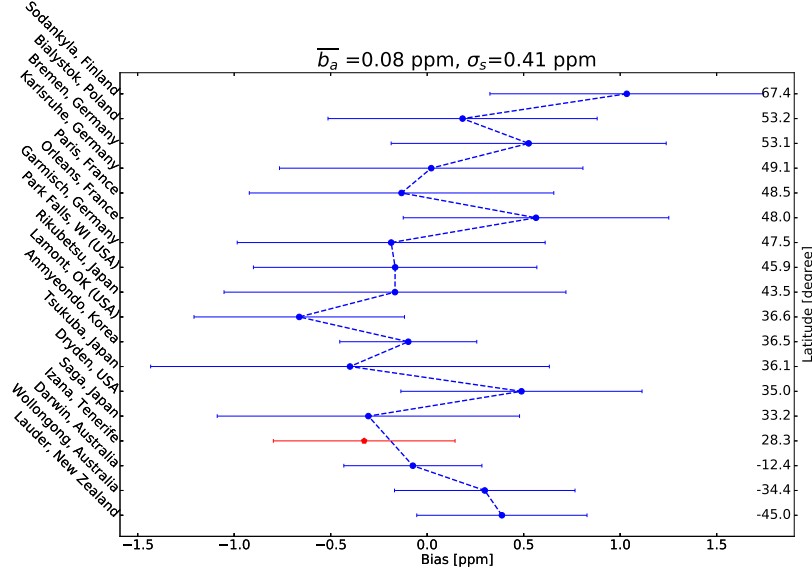

**Figure 9.** Same as Fig. 8, but for OCO-2 land type measurements obtained under nadir and glint modes.





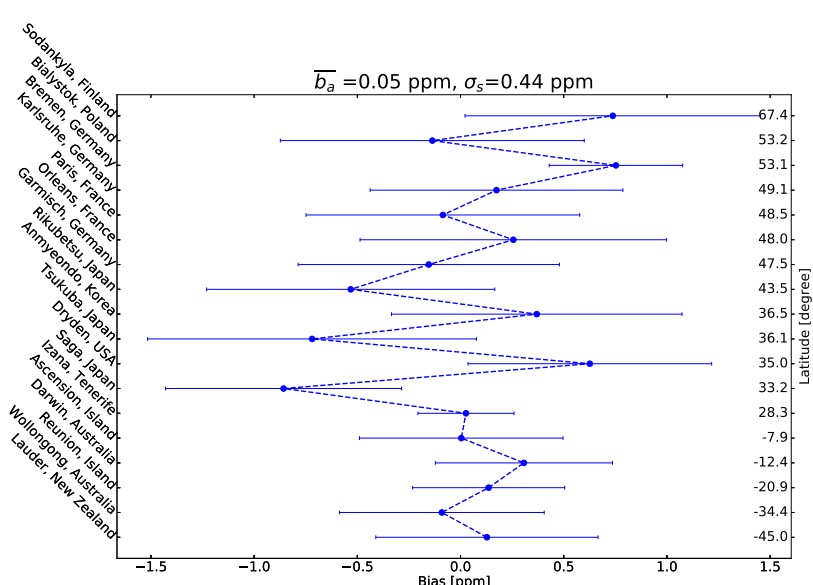

**Figure 10.** Same as Fig. 8, but for OCO-2 ocean type measurements obtained under glint mode.



**Table 2.** Bias between $X_{CO_2}$ retrieval from RemoTeC/OCO-2, including target, land and ocean retrievals, and TCCON data at individual stations in four different time intervals of a calendar year (Q1:1 January-31 March, Q2:1 April-30 June, Q3:1 July-30 September, Q4:1 October-31 December). For each time interval, we only use data from stations with more than 5 collocated points. In each table cell, bias, standard deviation and number of points are included and those with larger standard error ($\sigma/\sqrt{N} > 0.5$ ppm) after bias correction will also be neglected as done by Dils et al. (2014). For stations with all four seasonal biases, the standard deviation of these four biases ("Seas") are also caculated. This parameter is an indicator of their seaonal variability.

| Stations | Q1 | Q2 | Q3 | Q4 | Seas | Reference |
|---|---|---|---|---|---|---|
| Sodankyla, Finland (67.3N, 26.6E) | - | 0.70(1.49, 39) | 1.18(1.28, 30) | - | - | Kivi et al. (2014) |
| Bialystok, Poland (53.2N, 23.0E) | -0.34(1.34, 14) | 0.02(1.31, 40) | 0.62(1.60, 25) | 0.02(0.93, 7) | 0.34 | Deutscher et al. (2015) |
| Bremen, Germany (53.1N, 8.8E) | - | -0.04(0.95, 7) | 1.04(1.20, 14) | - | - | Notholt et al. (2014) |
| Karlsruhe, Germany (49.1N, 8.4E) | - | -0.16(1.37, 25) | 0.09(1.75, 24) | 0.59(0.75, 6) | - | Hase et al. (2015) |
| Park Falls, WI(USA) (48.4N, 2.3E) | -0.14(1.16, 17) | -0.37(1.53, 38) | 0.10(1.52, 46) | -0.44(1.27, 20) | 0.21 | Wennberg et al. (2014) |
| Paris, France (48.4N, 2.3E) | - | -0.15(1.10, 11) | 0.33(1.44, 19) | - | - | Te et al. (2014) |
| Izana, Tenerife (48.4N, 2.3E) | -0.24(0.73, 7) | - | - | - | - | Blumenstock et al. (2014) |
| Orleans, France (47.9N, 2.1E) | 0.36(1.01, 19) | 0.34(1.04, 34) | 0.32(1.81, 25) | 0.98(1.47, 15) | 0.28 | Warneke et al. (2014) |
| Garmisch, Germany (47.4N, 11.0E) | -0.04(1.47, 15) | -0.49(1.56, 28) | 0.02(1.34, 23) | - | - | Sussmann and Rettinger (2014) |
| Rikubetsu, Japan (43.4N, 143.7E) | -1.21(1.64, 11) | -0.13(1.64, 13) | 0.81(1.03, 6) | -0.24(1.04, 7) | 0.71 | Morino et al. (2016b) |
| Lamont, OK(USA) (36.6N, 97.4W) | -0.71(1.06, 55) | -0.35(1.01, 53) | -0.51(1.29, 59) | -1.00(0.83, 49) | 0.24 | Wennberg et al. (2016) |
| Anmyeondo, Korea (36.5N, 126.3E) | -0.26(0.58, 5) | - | 0.67(0.85, 7) | - | - | Goo et al. (2014) |
| Tsukuba, Japan (36.0N, 140.1E) | -1.31(1.18, 26) | 0.07(1.17, 12) | - | -1.00(1.17, 29) | - | Morino et al. (2016a) |
| Dryden, USA (34.9N, 117.8W) | 0.10(1.08, 40) | 0.85(0.99, 59) | 0.55(1.56, 48) | 0.16(1.24, 39) | 0.30 | Iraci et al. (2016) |
| Saga, Japan (33.2N, 130.2E) | -1.24(0.80, 14) | -0.93(1.05, 27) | -0.32(1.86, 24) | -0.19(1.33, 23) | 0.43 | Kawakami et al. (2014) |
| Ascension, Island (7.9165S, 14.3325W) | 0.19(1.03, 12) | 0.07(0.92, 18) | -0.04(0.99, 14) | -0.12(0.99, 23) | 0.12 | Feist et al. (2014) |
| Darwin, Australia (12.4S, 130.9E) | -0.21(0.88, 55) | 0.01(0.71, 61) | 0.38(0.58, 49) | 0.04(0.81, 66) | 0.21 | Griffith et al. (2014a) |
| Reunion, Island (20.901S, 55.485E) | 0.10(0.69, 9) | -0.23(0.75, 17) | 0.12(0.61, 25) | 0.50(0.73, 19) | 0.26 | De Mazière et al. (2014) |
| Wollongong, Australia (34.4S, 150.8E) | 0.04(0.98, 41) | 0.26(0.93, 17) | 0.21(1.18, 26) | 0.19(0.76, 37) | 0.08 | Griffith et al. (2014b) |
| Lauder, New Zealand (45.0S, 169.6E) | 0.19(0.99, 29) | 0.53(0.67, 10) | 0.13(0.92, 8) | 0.31(0.97, 37) | 0.15 | Sherlock et al. (2014) |
| ALL | 0.52 | 0.42 | 0.43 | 0.54 | - | |
| SRA | SRA=0.52 | | | | | |







**Figure 11.** Time variation of $X_{CO_2}$ difference between retrievals from OCO-2 observations over land (red dots) and ocean (blue pentagon) and collocated TCCON measurements for each TCCON station. Standard deviation of individual TCCON measurement and satellite retrievals are presented with the length of bar. In each subplot, the overall bias ($b$), standard deviation($\sigma$) and site location in latitude and longitude are included. The shown results here are bias-corrected data used in Table 2. An second order polynomial (blue dot lines) is fitted for distinguishing the time-variation of biases.





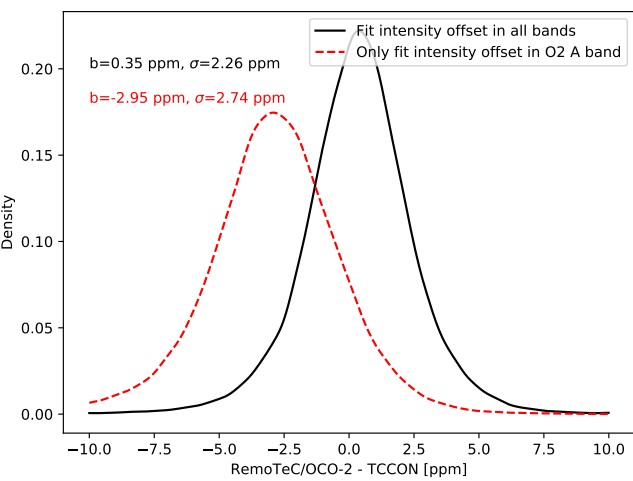

**Figure 12.** Density distributions of the $X_{CO_2}$ differences between OCO-2 land retrievals and collocated TCCON data for two different retrieval settings. In the black solid line we fit intensity offsets in all three OCO-2 bands while in the red dashed line we only fit the intensity offset in $O_2A$ band. Here we only do algorithm convergence filtering for both and take the intersection of them for fair comparison. The bias $b$ and sounding precision $\sigma$ for each retrieval are included.





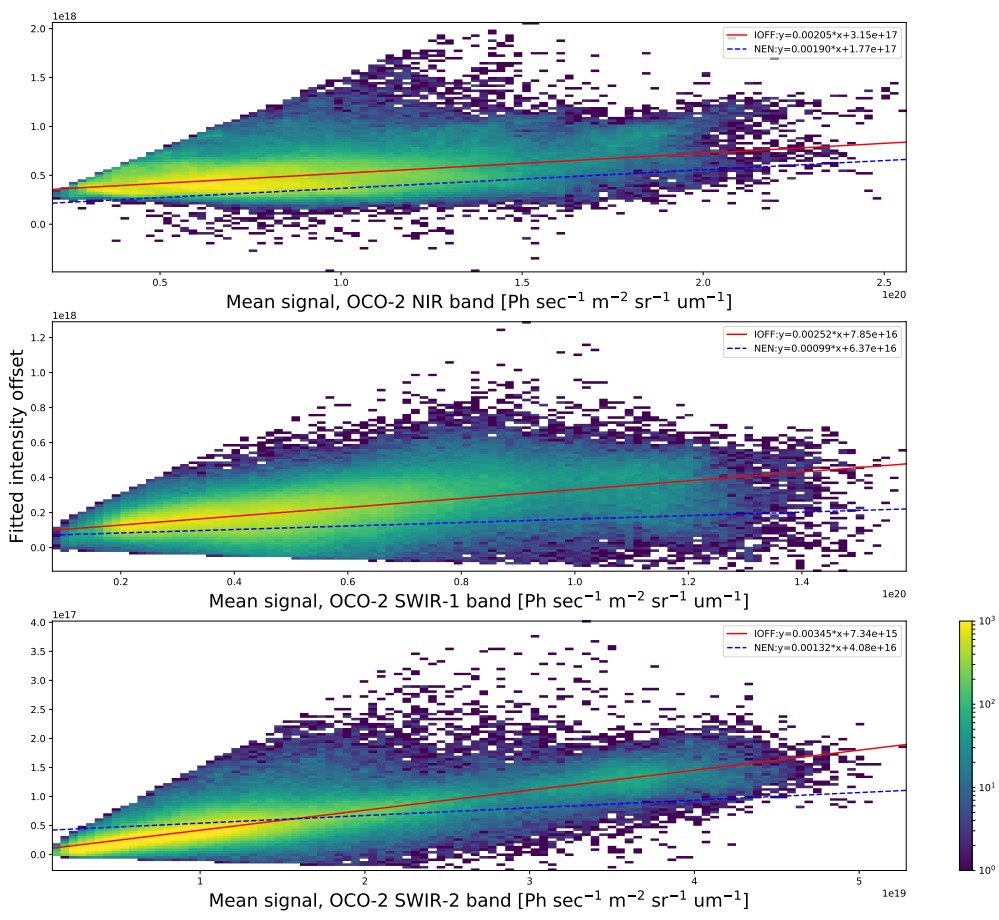

**Figure 13.** Variation of fitted intensity offset with respect to mean signals measured in each OCO-2 band for observations over land. Linear regression fit for the intensity offset (IOFF) and noise equivalent radiance (NEN) is overplotted along with fitted cofficients on top right. Different colors represent the frequency of point occurrence.