# Peer review of "Carbon dioxide retrieval from OCO-2 satellite observations using the RemoTeC algorithm and validation with TCCON measurements"

_Atmospheric Measurement Techniques, 2017_

## Referee Comment (RC1) · Anonymous Referee #3 · 27 Feb 2018

This paper describes the XCO2 retrieval from OCO-2 spectra with the RemoTeC algorithm for two years of data, focused on TCCON collocations for validation purposes. OCO-2 currently delivers the most accurate and the largest dataset of NIR / SWIR radiance measurements for XCO2 estimation. RemoTeC is widely acknowledged as a state-of-the-art retrieval algorithm, already successfully applied to GOSAT. The retrieval of OCO-2 with RemoTeC is therefore widely expected and this work largely deserves a dedicated publication.

The paper is well written and concise. The main properties of OCO-2 are clearly reminded. The main assumptions of RemoTeC are recalled, but I understand that a

detailed description of the algorithm requires to read references, which may be a weakness for the self consistency of the paper. Maybe the paper should be more precise about the modification of the algorithm for OCO-2. The methodology is based on the systematic comparison with several TCCON stations. This is a classical, rigorous and probably the most accurate strategy for XCO2 missions, as the column sensitivities are similar and strong efforts have been made to trace the TCCON network to the WMO XCO2 standard. Such validation work requires estimation of random error, global and regional biases, which can only be obtained at a reasonable cost with the large data set of the TCCON network. The choice of a period larger than 1 year is essential to remove the seasonal effects.

The main result is that the residuals biases of OCO-2 / RemoTeC with the TCCON global network is lower than 0.1 ppm in absolute value, and up to 1 ppm when looking at individual stations. These low values, of the same order as the OCO-2 L2, prove the quality of RemoTeC and its application to OCO-2. The remaining station to station biases are still high for the needs of the flux community, meaning that research must continue to improve the retrieval scheme and the understanding of the instrument (beyond the scope of this paper). The bias correction shows its efficiency to empirically reduce the biases, but the magnitude of the correction is still too high to give a solid confidence in the final value.

I have some questions and remarks that I would like the authors to address before publication. These will probably not require new calculations, but only precisions and additional materials. I will try to focus my questions on the application of RemoTeC to OCO-2 and not to the RemoTeC algorithm itself which was the subject of previous papers properly quoted. One of the drawbacks of this paper is that literature on this topic is large, and the reasons of some assumptions have now become implicit (and could sometimes deserve to be questioned once again). I also noticed that several results are given only in the text whereas they show be given dedicated figures or tables (see comments). This has to be corrected before publication. Finally, I was

sometimes lost in the different statistics indicator (target, land, ocean; all footprints or daily averages; global bias, station to station bias, standard deviation). A clearer presentation and interpretation of them would be welcome before publication.

MAJOR COMMENTS - Page 3 line 13: The objective of the study requires further justification than Âń to enhance the reliability and confidence of the data product Âż. What does this study aim at? To challenge the official OCO-2 Level 2 (L2)? To improve RemoTeC through its application to the new OCO-2 dataset, more accurate than GOSAT? Will a new OCO-2 / RemoTeC be proposed in the future?

- As already mentioned, there is a lack of description of the algorithm, largely given by references. This is however very important to understand the differences with the OCO-2 L2.

- p4 l12: 5° around a TCCON station is very large (~500km). In such an area the CO2 may not be considered as uniform. What is your justification? Did you make any error budget, any sensitivity study?

- p4 l16: Why do you restrict VZA to <30° and not to a larger value? Is there also a restriction on SZA? Table 1 gives some information but in contradiction for VZA, maybe because it only applies to land and glint? Please precise.

- p5 & 6: I think the paper deserves a table describing exactly the content of the state vector.

- p6 l13: the description of the cloud screening is too light, I understand it is a copy of what is done by OCO-2. Do you use the information from the OCO-2 pre-processing, or did you develop your own algorithm? Did you make any performance study, and associated XCO2 sensitivity study? 30% is higher than the performance reached by OCO-2 (for land and ocean).

- p6 l28: please explain the reason why you separate land and ocean evaluation. Is it based only the aerosol argument (p7 l19)? Was it decided from the OCO-2 feedback

? OCO-2 does also but with another separation between land nadir and land glint.

- p6 l33: the assumption that TCCON station to station variability is zero is very strong and may not be excluded when interpreting the results.

- p7 l8: you talk about retrieval uncertainties; these uncertainties may be instrument dependent. Compared to Butz et al 2011, Gueret et al 2013b, did you reconsider your filters for OCO-2?

- p7 l25: I don't understand why you say Âń we look for possible correlations of errors with instrumental, geophysical, meteorological and retrieved parameters Âż. Actually, here you do not look for such correlations (as would the OCO-2 Bias Correction do), you only calculate a regression with chi2, which is different. This is an original bias correction and, as far as I know, it is the first time it is applied. What made you adopt such methodology? To my mind its drawbacks are that you loose interesting spectral information about the residuals. An error in retrieved albedo may lead to a large chi2 whereas it has very limited impact on XCO2. An error in line-mixing may lead to a small chi2 residual but have a strong impact on XCO2. I clearly do not say the approach is wrong, but I think that it is new and should really deserve deep study. The shape of the chi2 spectra would deserve attempts of interpretation. Why would you have only to regress with chi2 in SWIR-1 and not in the other bands? Why would this bias correction be required only for land, not for oceans? The spectroscopy and the instrument are the same. You say in section 4.1 the aerosol contribution is weak in ocean glint measurement, that could be an explanation but aerosols are not the only source of bias.

- p22: figure 4 should exhibit a fit, be given for lands and oceans, and for the 3 spectral bands.

- p8 l9: you say some correlation with parameters of table 1 are reduced, you clearly have to present these correlations by a figure or table before and after bias correction. Otherwise we cannot accept such affirmation.

- p8 l19: please show fig 2 before and after bias correction. Giving a rough value in the text (∼0.1ppm) is not enough.

- p8 l23: please define your averaging. I understand that in fig 2 (no averaging), you plot every OCO-2 single footprint minus the TCCON of the area at the same time. I understand that in fig 5, 6, 7 you average every OCO-2 single footprint – TCCON at the same in a window 5°*5°*2h, is that true? Is it mean(OCO-2) – mean(TCCON)? You mention a Âń daily averaging Âż p8 l32, but this term is confusing because you may encounter several collocations with several TCCON during the same day. In such a case, are the data of different TCCON in the same average? If not, you should maybe talk about « overpass averaging »?

- p8 l23: Please explain why you make an averaging in 4.2 whereas you do not in 4.1. I guess that in 4.1 you need to keep the individual parameters for your bias correction derivation, but this clearly needs to be explained. For this reason, fig 2 and fig 5,6,7 cannot be directly compared, and that is why the effect of bias correction is difficult to assess.

- p9 l1 and fig 5,6,7: Please also give the figures before bias correction for fig 5,6,7, and give the associated standard deviation as for p8 l33.

- p9 l10: please illustrate the effect of bias correction by giving figure 8 also for before bias correction.

- p9: I know you made the assumption the TCCON stations are consistent (p6 l34), but I am disappointed that you do not try to interpret the station to station variability in terms of residual bias of OCO-2 / RemoTeC or actual differences between TCCON stations. The fact that you do not see the same station to station bias in land and oceans modes could suggest there are still biases in OCO-2 / RemoTeC.

- p10 l6: Did you try also the same retrieval as fig 6 without the fit of the offset in the O2 band? As you mention later, there could be a link with the stronger internal reflections

in this band.

- p10 l12: I am not convinced by the explanation of the lack of SIF fitting, since the behaviour of the SIF and the offset is very different (SIF exhibits atmospheric absorptions).

- p10 section 5: I think the comparison with the previous OCO-2 / ACOS – TCCON and with the previous GOSAT / RemoTeC – TCCON are very important. But here the discussion is poor, mentioning only the common results in terms of standard deviation. This section really deserves to compare biases (global, station to station, etc.), as it was the case for section 4. This could help understand the origin of biases (from TCCON, from the instrument, the retrieval code). This should be done before and after bias correction.

MINOR COMMENTS - p2 l9: for clarity I would make a new paragraph.

- p4 l3: please precise if you use the tabulated instrumental functions given in the OCO-2 products.

- p4 l12: you use a requirement in degrees rather than in km, why? This makes a distance criterion in km variable according to the station, which is not suitable.

- p4 l18: please precise is you make your own calculation of surface pressure from ECMWF and MNT (this information could also be read from the OCO-2 L2 data). How do you interpolate ECMWF and how to select SRTM grid points? This question makes sense since you do not retrieve surface pressure in your state vector.

- p4 l23: Your initial guess for CO2 and CH4 comes from different years, therefore it is subject to inter-annual variability. What is the sensitivity of your retrieval to this first guess?

- p6 l24: figure 1 is given as an illustration but is not interpreted. It is very quickly mentioned on p9 l22, but not comparable. Please emphasize the scientific value of the figure, or discard it.

- p6 l6: by « second order spectral dependence of the Lambertian surface albedo »,
do you mean you retrieve 3 coefficients of a polynomial describing the albedo? Please
precise.

- p6 l31: the « SRA » clearly deserves a mathematical definition, and not just a
reference to Dils et al.

- p7 l1: « however » is inappropriate (before you discussed station to station bias, after
you discuss year to year variability and global uncertainty.

- p7 l3: the interest of a validation with TCCON deserves a stronger argumentation
(see my introduction).

- p7 l24: For clarity I would cut here and create a new paragraph: one for « Filters »,
one for « Bias correction », since these are 2 different topics.

- p7 l15: how do you define exactly delta_i ?

- p8 l7: is chi2 the chi2 of the SWIR-1 band, as suggested by the above text? Please
precise in the formula.

- p22: legend should precise it is land.

- p8 l18: avoid the terms « precisions » and « accuracy » unless you unambiguously
define them. Prefer bias and random error.

- p9 l4 and figure 8,9,10: please give the number of collocation per station.

- p21: fig 2 has 3 panels, fig 5,6,7 could also be the 3 panels of a single figure. Please
harmonize.

- p10 l9: please explain why you compare the slope of retrieved offset with that of noise.
An offset may be related to the signal (for example straylight) or not (for example wrong
dark current estimation).

- p10 l11: why do you say the offset in the O2 band shows a less strong dependence

on the signal? The slope is the same as in the other bands (there is only a difference in the noise slope).

- fig8,9,10: giving the actual value of bias and standard deviation for each station would be of great value.

- p11 l14: « a posteriori » is a bit misleading, since at least SZA, VZA and sev are probably pre-filtered.

- p11 l22: you compare target and lands without reminding the values. I guess you talk about station to station variability given by figures 8 and 9 (0.35 for target and 0.41 for land), and not the global biases given by figures 5 and 6 (-0.07 for target and 0.00 for land). This should be recalled.

---

## Referee Comment (RC3) · C. O'Dell (Referee) · 8 Mar 2018

Review of "Carbon dioxide retrieval from OCO-2 satellite observations using the RemoTeC algorithm and validation with TCCON measurements"

by Wu et al.
*Atmos. Meas. Tech. Disc.*, amt-2017-415

This paper is describes the relatively straightforward application of the RemoTeC retrieval algorithm to data from the Orbiting Carbon Observatory-2 (OCO-2). RemoTeC was written to retrieve $CO_2$ and $CH_4$ column-average concentrations from the Greenhouse Gases Observing Satellite (GOSAT), and has been improved and validated over the years, as described in a number of publications. The authors have applied this mature algorithm to OCO-2 data to retrieve $CO_2$ (OCO-2 does not have the $CH_4$ band that GOSAT possesses), and find that after a few slight modifications, the error statistics of OCO-2 retrievals vs. ground truth data compare favorably both to the operational OCO-2 product as well as to RemoTeC retrievals of $CO_2$ from GOSAT.

*General Comments*

The paper is useful in that it shows that the RemoTeC algorithm can be successfully applied to OCO-2, though it is relatively dry and offers few new physical insights into sources of error/bias in the OCO-2 measurements. However, is it worthwhile piece of work, and I recommend publication after making some minor revisions.

My only main comment on the paper has to do with the filtering and bias correction, for which the bottom-line recipes are given. Some more information would be welcome. For instance, what other parameters were investigated for bias correction or filtering, such as the 1/(size parameter) variable used in GOSAT bias correction (Guerlet et al, 2013b)? Was the $\omega_s$ parameter of Guerlet et al. (2013b) found not to be useful for OCO-2, even though it was for GOSAT? A figure similar to that of Figure 11 in Guerlet et al. (2013b) would be very useful here to see how similar/different GOSAT vs. OCO-2 retrieval biases are. Also, how stringent were your filters overall – did you filter out 10% L2-processed soundings, 50%, etc? How was this different over land and ocean? A throughput map would be useful.

*Specific Comments (P=page, L=line)*
P2, L24: "XCO2 retrievals with this level of accuracy [<1%] can provide valuable information on…sources and sinks…" No! 1% = 4 ppm. We know that regional biases even 1 ppm (0.25%) in XCO2 are too large (Chevallier et al, 2014). Please modify or remove this statement.

P3, L2: "by aerosols and cirrus." Do water clouds not have any effect on scattering? I suggest changing this statement to "by aerosols and clouds."

P3, 1st paragraph. The authors describe a number of XCO2 retrieval algorithms but this list is certainly not exhaustive. There are the BESD and FOCAL retrievals from

M. Reuter, the TanSAT retrieval from D. Yang, and various versions of the PPDF retrieval of Oshchepkov and Bril. You should either cite these or make clear that you are not exhaustively listing all available retrievals.

P4, L22: "barometric law". Do you not mean the hypsometric equation (which combines the ideal gas law with the hydrostatic equation). They may be equivalent, I'm not entirely sure. But usually in this context, it is referred to as the hypsometric equation.

P4, L23: Are your priors adjusted for the secular growth rate of CO2 (since you just say you use CT from 2013)? Seems like you should, or you could probably introduce an artificial trend in your retrievals.

P5, L20: Does $\mathbf{S_y}$ include any estimate of forward model error, as you previously implied it might (P5, L2). Similar are the noise estimates taken from the OCO-2 suggested formulation, or do you calculate your own noise estimates somehow?

P6, L9-10: Please discuss whether the per-band radiance offsets were needed for GOSAT. My understanding is that they were needed for band 1, but not the other two bands. You could instead bring this up in section 4.3 as well, but I think it's important to contrast this need for the offsets in OCO-2 vs. that of GOSAT. For instance – I was thinking that maybe you needed them because your retrieval doesn't explicitly retrieve cirrus, so it would be difficult to retrieval soundings with a cirrus layer overlying a thin aerosol layer, which is a pretty common situation. Retrieval the 3 per-band offsets would be a pretty easy way to fake it. But that would likely then also be needed for GOSAT. Some discussion on this would be useful.

P7, L14: $\chi_2$ is the symbol usually referred to as the total chi-squared. What you show is much more similar to the "reduced chi-squared", which is the total chi-squared divided by the number of degrees of freedom (# channels - # retrieved parameters). You really are giving the mean chi-squared per channel. You should make this clear, and that a value around unity would indicate a fit that is in line with the noise. Values consistently higher than unity mean there are the systematic errors in the forward model that are not able to be fitted away.

P8, top: In the discussion of using the SWIR-1 chi-squared as a bias correction parameter, it would be nice to lengthen this discussion. Does SWIR-2 chi-squared perform similarly? Other parameters? Mention that r=0.2 means that 0.04 percent of the variance is explained (or it will reduce the standard deviation by about 2%). Why do you include the offset "d" parameter when you already include a global bias correction? They would be directly related to each other. Does this multiplicative formula (equation 4) work better than an additive equation? Finally, it would be valuable if you could speculate on *why* this parameter seems to be correlated with the bias over land. And perhaps on why it is NOT correlated with the bias over water. Are the chi-squared values much lower over water? Finally, what is the

spatial distribution of this parameter?  Is it highly scattered or does it seem to remove coherent regional biases?

P9, L10: Some comments on why the effect of the bias correction is largest for those 3 stations would be welcome.  It seems like it should be substantial for all over-land stations, unless the chi-squared values were just worse for those stations.  My guess is that your chi-squared is going to be correlated with SNR or surface albedo, and brighter surfaces will have larger corrections.  If you plotted the mean correction on a map, this would probably become obvious.

Section 4.3 As mentioned before, contrast with the offset approach for GOSAT.  Is the behavior of the fitted radiance offsets similar over land and ocean?  How correlated are the fitted offsets for the 3 bands? (either in absolute terms, or relative to the mean radiance in their respective bands)  If they are highly correlated, or not, that would give you a clue what they are correcting for (either cirrus, as I hypothesized earlier, or some instrument effect that is particular to OCO-2, and perhaps not GOSAT).

P11, top: Are the GOSAT vs. OCO-2 error statistics vs. TCCON similar for both land and ocean soundings?

Figures: in many of the figures, the font sizes make reading some of the text difficult (axis labels, bias numbers , TCCON site names, etc).  Please try to make them bigger to increase legibility.

*Technical comments*
P4, various: spectral samplings → spectral samples
P5, L10: "radiative transfer model Hasekamp…" → "radiative transfer model (Hasekamp…"
P10, L8: proportional mis-spelled

---

## Author Comment (AC1) · 1 May 2018

We thank all reviewers for their constructive comments, which helped to improve the paper. Below, we address all comments point-by-point.

**#reviewer1**

This paper describes the XCO2 retrieval from OCO-2 spectra with the RemoTeC algorithm for two years of data, focused on TCCON collocations for validation purposes. OCO-2 currently delivers the most accurate and the largest dataset of NIR / SWIR radiance measurements for XCO2 estimation. RemoTeC is widely acknowledged as a state-of-the-art retrieval algorithm, already successfully applied to GOSAT. The retrieval of OCO-2 with RemoTeC is therefore widely expected and this work largely deserves a dedicated publication. The paper is well written and concise. The main properties of OCO-2 are clearly reminded. The main assumptions of RemoTeC are recalled, but I understand that a detailed description of the algorithm requires to read references, which may be a weakness for the self consistency of the paper. Maybe the paper should be more precise about the modification of the algorithm for OCO-2. The methodology is based on the systematic comparison with several TCCON stations. This is a classical, rigorous and probably the most accurate strategy for XCO2 missions, as the column sensitivities are similar and strong efforts have been made to trace the TCCON network to the WMO XCO2 standard. Such validation work requires estimation of random error, global and regional biases, which can only be obtained at a reasonable cost with the large data set of the TCCON network. The choice of a period larger than 1 year is essential to remove the seasonal effects.

The main result is that the residuals biases of OCO-2 / RemoTeC with the TCCON global network is lower than 0.1 ppm in absolute value, and up to 1 ppm when looking at individual stations. These low values, of the same order as the OCO-2 L2, prove the quality of RemoTeC and its application to OCO-2. The remaining station to station biases are still high for the needs of the flux community, meaning that research must continue to improve the retrieval scheme and the understanding of the instrument (beyond the scope of this paper). The bias correction shows its efficiency to empirically reduce the biases, but the magnitude of the correction is still too high to give a solid confidence in the final value.

I have some questions and remarks that I would like the authors to address before publication. These will probably not require new calculations, but only precisions and additional materials. I will try to focus my questions on the application of RemoTeC to OCO-2 and not to the RemoTeC algorithm itself which was the subject of previous papers properly quoted. (1)-One of the drawbacks of this paper is that literature on this topic is large, and the reasons of some assumptions have now become implicit (and could sometimes deserve to be questioned once again). (2)-I also noticed that several results are given only in the text whereas they show be given dedicated figures or tables (see comments). This has to be corrected before publication. (3)-Finally, I was sometimes lost in the different statistics indicator (target, land, ocean; all footprints or daily averages; global bias, station to station bias, standard deviation). A clearer presentation and interpretation of them would be welcome before publication.

C1-Page 3 line 13: The objective of the study requires further justification than « to enhance the reliability and confidence of the data product ». What does this study aim at? To challenge the official OCO-2 Level 2 (L2)? To improve RemoTeC through its application to the new OCO-2 dataset, more accurate than GOSAT? Will a new OCO-2 / RemoTeC be proposed in the future?

R1-. We added a phrase "We expect that application of RemoTeC to OCO-2 data will lead to a better understanding to the capabilities and limitations of the OCO-2 instrument and the operational level-2 data product. Furthermore, we see this work as a first step towards processing a larger data set with RemoTeC." For example, in this paper we show that RemoTeC the bias correction only has a minor effect on the XCO2 retrieval accuracy, while for the official level-2 product a much larger correction is needed. This suggests that the need for bias correction is for large part caused by the algorithm itself rather than by instrument

related errors. Also we show that it is needed to fit an intensity offset which gives insight into instrumental errors of the OCO-2 instrument.

C2-As already mentioned, there is a lack of description of the algorithm, largely given by references. This is however very important to understand the differences with the OCO-2 L2.

R2- One major modification on RemoTeC/OCO-2 is that now we adopted a vector radiative transfer model (LINTRAN V2) to the retrieval scheme. Scattering is considered for ocean glint retrievals. Before, in GOSAT application, RemoTeC uses a scalar radiative transfer model for land and performs non-scattering retrieval for ocean glint. We add " For OCO-2 application, several modifications have been made to the algorithm: (1) a vector radiative transfer model (LINTRAN V2) is employed in the retrieval scheme; (2) Aerosol scattering effects are taken into account for ocean glint retrievals; (3) Information on pressure profiles, humidity and temperature are extracted from the ECMWF data with a resolution of 0.125 by 0.125 instead of 0.75 by 0.75 previously. ".

C3- p4 I12: 5° around a TCCON station is very large (~500km). In such an area the CO2 may not be considered as uniform. What is your justification? Did you make any error budget, any sensitivity study?

R3- The CO2 may be inhomogeneous in this area. However, in the validation we do not see a clear dependency between  $XCO_2$  difference and the collocation distance, as shown in Figure 1. We add "Here, the dependency of difference with collocation distance and surface pressure is negligible." in section 4.1 in the paper.

Figure 1. Dependency between XCO2 difference and TCCON distance (unit degree).

C4-p4 I16: Why do you restrict VZA to

Figure 2. Dependency between XCO2 difference and prior variation (unit ppm).

ref1: Butz, A., Guerlet, S., Hasekamp, O., Schepers, D., Galli, A., Aben, I., Frankenberg, C., Hartmann, J., Tran, H., Kuze, A., Keppel-Aleks, G., Toon, G., Wunch, D., Wennberg, P., Deutscher, N., Griffith, D., Macatangay, R., Messerschmidt, J., Notholt, J. and Warneke, T. (2011). Toward accurate CO2 and CH4 observations from GOSAT. Geophysical Research Letters, 38 (14), 1-6.

**Supporting Information for''Carbon dioxide retrieval from OCO-2 satellite observations using the RemoTeC algorithm and validation with TCCON measurements''**

Lianghai Wu1, Otto Hasekamp1, Haili Hu1, Jochen Landgraf1, Andre Butz2,3, Joost aan de Brugh1,

- 5 Ilse Aben1, Dave F. Pollard4, David W. T. Griffith5, Dietrich G. Feist6, Dmitry Koshelev7, Frank Hase8, Geoffrey C. Toon9, Hirofumi Ohyama10, Isamu Morino10, Justus Notholt 11, Kei Shiomi12, Laura Iraci13, Matthias Schneider15, Martine de Maziére 14, Ralf Sussmann15, Rigel Kivi16, Thorsten Warneke11, Tae-Young GOO17, and Yao Té7
- 1SRON Netherlands Institute for Space Research, Utrecht, The Netherlands
   2Institute of Atmospheric Physics, Deutsches Zentrum f
  ür Luft- und Raumfahrt e.V. (DLR), Wessling-Oberpfaffenhofen, Germany
   3Meteorologisches Institut, Ludwig-Maximilians-Universität (LMU), Munich, Germany
   4National Institute of Water and Atmospheric Research Ltd (NIWA), Lauder, New Zealand
- 5University of Wollongong, Wollongong, Australia
   6Max Planck Institute for Biogeochemistry, Jena, Germany
   7LERMA-IPSL, Sorbonne Universités, UPMC Univ Paris 06, CNRS, Observatoire de Paris, PSL Research University,
   75005,
- Paris, France
- 20 8Karlsruhe Institute of Technology (KIT), IMK-ASF, Karlsruhe, Germany
   9Jet Propulsion Laboratory, California Institute of Technology, Pasadena, California, USA
   10National Institute for Environmental Studies (NIES), Tsukuba, Japan
   11University of Bremen, Bremen, Germany
   12Japan Aerospace Exploration Agency, Tsukuba, Japan
   25 12NASA Amag Basagenth Center, Moffatt Field, CA, USA
- 13NASA Ames Research Center, Moffett Field, CA, USA
   14Royal Belgian Institute for Space Aeronomy, Brussels, Belgium
   15Karlsruhe Institute of Technology (KIT), Institute of Meteorology and Climate Research (IMK-IFU), Garmisch
   Partenkirchen, Germany
   16Finnish Meteorological Institute, Sodankylä, Finland
- 30 17National Institute of Meteorological Research, Seoul, Republic of Korea

Correspondence to: Lianghai Wu (l.wu@sron.nl)

**1** Overview**

- 5 Here, we provide additional information about: Fig.S1: Error on XCO2 retrievals as a function of six parameters: air mass, water column, blended albedo, mean signal in O2 A-band, aerosol ratio and aerosol size parameter (reff); Fig.S2: same as Fig. S1 but after bias correction; Fig.S3:Validation of individual XCO2 retrieved from OCO-2 measurements after bias correction; Fig.S4-Fig.S6: Validation of overpass averaged retrievals with TCCON before bias correction for targer, land and ocean soundings, respectively; Fig.S7-Fig.S9: The dependence of the bias on latitude before bias correction for targer, land
- 10 and ocean soundings, respectively.

**2 Content of this file**

(1). Figures S1 to S9.

(2). Table S1

15

---

## Author Comment (AC2) · 1 May 2018

We thank all reviewers for their constructive comments, which helped to improve the paper. Below, we address all comments point-by-point.

**reviewer2**

This paper is describes the relatively straightforward application of the RemoTeC retrieval algorithm to data from the Orbiting Carbon Observatory-2 (OCO-2). RemoTeC was written to retrieve CO 2 and CH 4 column-average concentrations from the Greenhouse Gases Observing Satellite (GOSAT), and has been improved and validated over the years, as described in a number of publications. The authors have applied this mature algorithm to OCO-2 data to retrieve CO 2 (OCO-2 does not have the CH 4 band that GOSAT possesses), and find that after a few slight modifications, the error statistics of OCO-2 retrievals vs. ground truth data compare favorably both to the operational OCO-2 product as well as to RemoTeC retrievals of CO2 from GOSAT.

C-General Comments

The paper is useful in that it shows that the RemoTeC algorithm can be successfully applied to OCO-2, though it is relatively dry and offers few new physical insights into sources of error/bias in the OCO-2 measurements. However, is it worthwhile piece of work, and I recommend publication after making some minor revisions. My only main comment on the paper has to do with the filtering and bias correction, for which the bottom-line recipes are given. Some more information would be welcome. For instance, what other parameters were investigated for bias correction or filtering, such as the 1/(size parameter) variable used in GOSAT bias correction (Guerlet et al, 2013b)? Was the ω s parameter of Guerlet et al. (2013b) found not to be useful for OCO-2, even though it was for GOSAT? A figure similar to that of Figure 11 in Guerlet et al. (2013b) would be very useful here to see how similar/different GOSAT vs. OCO-2 retrieval biases are. Also, how stringent were your filters overall –did you filter out 10% L2-processed soundings, 50%, etc? How was this different over land and ocean? A throughput map would be useful.

R-The correlations of XCO2 difference with other parameters as used for GOSAT are now shown in Figure S1 in attachment. We tried the potential bias correction parameter (aerosol size, reff) as previously used by GOSAT retrievals. However, as can be seen in the bottom right panel of **Fig.S1** in 'Supporting Information' (SI)  there is no clear dependency between this parameter.

For the overall throughput, we add in the paper "The overall L2-processed throughput is around 15%. When estimated separately, the percentages are 15.8%, 14.0% and 16.0% for target, land and ocean soundings, respectively."

C1- P2, L24: "XCO2 retrievals with this level of accuracy [<1%] can provide valuable information on...sources and sinks..." No! 1% = 4 ppm. We know that regional biases even 1 ppm (0.25%) in XCO2 are too large (Chevallier et al, 2014). Please modify or remove this statement.

R1-Modified. The statement now becomes " The XCO2 derived from GOSAT has an accuracy in the order of a few tents of a percent. XCO2 retrievals with this level of accuracy can provide valuable information on the variation of CO2 ."

C2-P3, L2: "by aerosols and cirrus." Do water clouds not have any effect on scattering? I suggest changing this statement to "by aerosols and clouds."
R2-modified.

C3-P3, 1 st paragraph. The authors describe a number of XCO2 retrieval algorithms but this list is certainly not exhaustive. There are the BESD and FOCAL retrievals from M. Reuter, the TanSAT retrieval from D.

Yang, and various versions of the PPDF retrieval of Oshchepkov and Bril. You should either cite these or make clear that you are not exhaustively listing all available retrievals.

R3- Indeed, we are not trying to list all available retrieval algorithms so we use the word 'including'.

C4- P4, L22: "barometric law". Do you not mean the hypsometric equation (which combines the ideal gas law with the hydrostatic equation). They may be equivalent, I'm not entirely sure. But usually in this context, it is referred to as the hypsometric equation.
R4- Modified.

P4, L23: Are your priors adjusted for the secular growth rate of CO2 (since you just say you use CT from 2013)? Seems like you should, or you could probably introduce an artificial trend in your retrievals.
R5- No, for now, the growth rate is not considered in the prior. It is important to note that the retrieval results are hardly affected by the prior, see Figure 1.

[Figure]

Figure 1. Dependency between XCO$_2$ difference and prior variation (unit ppm).

P5, L20: Does Sy include any estimate of forward model error, as you previously implied it might (P5, L2). Similar are the noise estimates taken from the OCO-2 suggested formulation, or do you calculate your own noise estimates somehow?
R6-Sy only include noise estimates from the OCO-2 suggested formulation. Modified to be clear.

P6, L9-10: Please discuss whether the per-band radiance offsets were needed for GOSAT. My understanding is that they were needed for band 1, but not the other two bands. You could instead bring this up in section 4.3 as well, but I think it's important to contrast this need for the offsets in OCO-2 vs. that of GOSAT. For instance – I was thinking that maybe you needed them because your retrieval doesn't explicitly retrieve cirrus, so it would be difficult to retrieval soundings with a cirrus layer overlying a thin aerosol layer, which is a pretty common situation. Retrieval the 3 per-band offsets would be a pretty easy way to fake it. But that would likely then also be needed for GOSAT. Some discussion on this would be useful.
R7- For GOSAT the intensity offset fitting is only needed for the O2A band but not for the SWIR bands. This suggests that the offset corrects for an instrumental artefact rather than a retrieval artefact. The state vector differences between OCO-2 and GOSAT are now added in 4.3 section.

P7, L14: χ2 is the symbol usually referred to as the total chi-squared. What you show is much more similar to the "reduced chi-squared", which is the total chi- squared divided by the number of degrees of freedom (# channels - # retrieved parameters). You really are giving the mean chi-squared per channel. You should make this clear, and that a value around unity would indicate a fit that is in line with the noise. Values consistently higher than unity mean there are the systematic errors in the forward model that are not able to be fitted away.

R8- Indeed, χ2 is the "reduced chi-squared ". Modified accordingly in the paper.

P8, top: In the discussion of using the SWIR-1 chi-squared as a bias correction parameter, it would be nice to lengthen this discussion. Does SWIR-2 chi-squared perform similarly? Other parameters? Mention that r=0.2 means that 0.04 percent of the variance is explained (or it will reduce the standard deviation by about 2%). Why do you include the offset "d" parameter when you already include a global bias correction? They would be directly related to each other. Does this multiplicative formula (equation 4) work better than an additive equation? Finally, it would be valuable if you could speculate on why this parameter seems to be correlated with the bias over land. And perhaps on why it is NOT correlated with the bias over water. Are the chi-squared values much lower over water? Finally, what is the spatial distribution of this parameter? Is it highly scattered or does it seem to remove coherent regional biases?

R9- For bias correction, using SWIR-2 chi-squared gives similar results as using SWIR-1 chi-squared when we look at some overall statistics like station-to-station bias, or standard deviation. Using other parameters like those listed in Figure S1, the performance is different and can increase the station-to-station bias. Indeed, the goal of "d" in the bias correction is to correct a global bias. We tried to keep the bias correction purely multiplicative, since the leading scaling term would just link the spectroscopic calibration to the in-situ calibration. The performance is more or less the same with an additive equation.

The reason why the SWIR-1 chi-squared is highly correlated with the bias over land but not with that over ocean is probably related to the fact that high chi2 values over land are often related to bright surfaces. We know that retrieving aerosol over bright land surface is challenging. What we also see here is that SWIR-1 chi-squared is highly correlated (cor is around 0.75) with land surface albedo. However, using albedo directly to do the bias correction can NOT achieve similar performance and will make some statistics worse, for example seasonal variations. So, apart from bright surfaces, the bias correction with chi2 corrects XCO2 retrievals for cases where the forward model is less capable of fitting the measurements. By doing the correction, as we can see in the paper, we can reduce regional biases since the station-to-staion bias become less. For ocean glint, as we mentioned, aerosols play a less important role and mainly act as an extinction layer. Thus, we cannot see similar feature with land retrievals.

P9, L10: Some comments on why the effect of the bias correction is largest for those 3 stations would be welcome. It seems like it should be substantial for all over-land stations, unless the chi-squared values were just worse for those stations. My guess is that your chi-squared is going to be correlated with SNR or surface albedo, and brighter surfaces will have larger corrections. If you plotted the mean correction on a map, this would probably become obvious.

R10- Indeed, the chi-squared is highly correlated (correlation coefficient is around 0.75) with surface albedo. This could be partly attributed to aerosols since it is difficult to account scattering effects of aerosols over bright surfaces. We added a comment with the reason for the large correction effect for the 3 stations, " This happens due to that the goodness of fit is highly correlated with surface albedo and thus make the corrections apparently to regions with large albedos.".

Section 4.3 As mentioned before, contrast with the offset approach for GOSAT. Is the behavior of the fitted radiance offsets similar over land and ocean? How correlated are the fitted offsets for the 3 bands? (either in absolute terms, or relative to the mean radiance in their respective bands) If they are highly correlated, or not, that would give you a clue what they are correcting for (either cirrus, as I hypothesized earlier, or some instrument effect that is particular to OCO-2, and perhaps not GOSAT).

R11- The fitted radiance offsets are similar in retrievals over land and ocean. The intensity offsets for the 3 bands are moderately correlated with each other (around 0.35).

P11, top: Are the GOSAT vs. OCO-2 error statistics vs. TCCON similar for both land and ocean soundings?

R12- It is difficult to compare OCO-2 ocean retrievals to that of GOSAT because the collocated TCCON sites are quite limited (4 stations).

Figures: in many of the figures, the font sizes make reading some of the text difficult (axis labels, bias numbers , TCCON site names, etc). Please try to make them bigger to increase legibility.

R13-Thanks for the suggestion. We updated this in the revised manuscript.

Technical comments
P4, various: spectral samplings à spectral samples
P5, L10: "radiative transfer model Hasekamp..." à "radiative transfer model (Hasekamp..."
P10, L8: proportional mis-spelled

R14- modified.

**Supporting Information for"Carbon dioxide retrieval from OCO-2 satellite observations using the RemoTeC algorithm and validation with TCCON measurements"**

Lianghai Wu[1], Otto Hasekamp[1], Haili Hu[1], Jochen Landgraf[1], Andre Butz[2,3], Joost aan de Brugh[1],
Ilse Aben[1], Dave F. Pollard[4], David W. T. Griffith[5], Dietrich G. Feist[6], Dmitry Koshelev[7], Frank
Hase[8], Geoffrey C. Toon[9], Hirofumi Ohyama[10], Isamu Morino[10], Justus Notholt [11], Kei Shiomi[12],
Laura Iraci[13], Matthias Schneider[15], Martine de Maziére [14], Ralf Sussmann[15], Rigel Kivi[16], Thorsten
Warneke[11], Tae-Young GOO[17], and Yao Té[7]

[1]SRON Netherlands Institute for Space Research, Utrecht, The Netherlands
[2]Institute of Atmospheric Physics, Deutsches Zentrum für Luft- und Raumfahrt e.V. (DLR), Wessling-Oberpfaffenhofen,
Germany
[3]Meteorologisches Institut, Ludwig-Maximilians-Universität (LMU), Munich, Germany
4National Institute of Water and Atmospheric Research Ltd (NIWA), Lauder, New Zealand
5University of Wollongong, Wollongong, Australia
6Max Planck Institute for Biogeochemistry, Jena, Germany
7LERMA-IPSL, Sorbonne Universités, UPMC Univ Paris 06, CNRS, Observatoire de Paris, PSL Research University,
75005,
Paris, France
8Karlsruhe Institute of Technology (KIT), IMK-ASF, Karlsruhe, Germany
9Jet Propulsion Laboratory, California Institute of Technology, Pasadena, California, USA
10National Institute for Environmental Studies (NIES), Tsukuba, Japan
11University of Bremen, Bremen, Germany
12Japan Aerospace Exploration Agency, Tsukuba, Japan
13NASA Ames Research Center, Moffett Field, CA, USA
14Royal Belgian Institute for Space Aeronomy, Brussels, Belgium
15Karlsruhe Institute of Technology (KIT), Institute of Meteorology and Climate Research (IMK-IFU), Garmisch
Partenkirchen, Germany
16Finnish Meteorological Institute, Sodankylä, Finland
17National Institute of Meteorological Research, Seoul, Republic of Korea

*Correspondence to*: Lianghai Wu (l.wu@sron.nl)

**1 Overview**

Here, we provide additional information about: Fig.S1: Error on XCO2 retrievals as a function of six parameters: air mass, water column, blended albedo, mean signal in O2 A-band, aerosol ratio and aerosol size parameter (reff); Fig.S2: same as Fig. S1 but after bias correction; Fig.S3:Validation of individual XCO2 retrieved from OCO-2 measurements after bias correction; Fig.S4-Fig.S6: Validation of overpass averaged retrievals with TCCON before bias correction for targer, land and ocean soundings, respectively; Fig.S7-Fig.S9: The dependence of the bias on latitude before bias correction for targer, land and ocean soundings, respectively.

**2 Content of this file**

(1). Figures S1 to S9.

(2). Table S1

[Figure]

Figure S1. Error on XCO2 retrievals as a function of six parameters: air mass, water column, blended albedo, mean signal in O2 A-band, aerosol ratio and aerosol size parameter (reff). Different colors represent the frequency of point occurrence.

[Figure]

Figure S2.Same as Figure S1 after bias correction.

[Figure]

5    Figure S3. Validation of individual XCO2 retrieved from OCO-2 measurements with collocated TCCON data after bias correction.

[Figure]

Figure S4. Validation of averaged XCO2 retrieved from OCO-2 target measurements with collocated TCCON data before bias correction. The standard deviation of individual TCCON data and that of RemoTeC/OCO-2 retrievals are presented with error bars. The bias ($b_a$), standard deviation ($\sigma_a$), number of points (N), the Pearson correlation coefficient (cor) and one-to-one line are included.

[Figure]

Figure S5. Same as Fig. S4, but for OCO-2 land type measurements obtained under nadir and glint modes.

[Figure]

Figure S6. Same as Fig. S4, but for OCO-2 ocean type measurements obtained under glint mode.

[Figure]

Figure S7. The dependence of the bias between RemoTeC/OCO-2 target XCO$_2$ retrievals coincident with TCCON data on the latitude of each station. Shown are the averaged results before bias correction. Stations with less than 5 collocation points (marked with red pentagon) should be interpreted with care and are therefore excluded from the calculation of the derived parameters including mean bias (b$_a$) and the station-to-station variability (σ$_s$). The size of each dot represents the standard deviation of the difference at each station.

[Figure]

Figure S8. Same as Fig. S7, but for OCO-2 land type measurements obtained under nadir and glint modes.

[Figure]

Figure S9. Same as Fig. S7, but for OCO-2 ocean type measurements obtained under glint mode.

Table S1. Correlation coefficients between XCO2 difference between TCCON and OCO-2 retrievals with filtering parameters listed in Table 2 in the paper.

| parameters | Correlation before bias correction | Correlation after bias correction |
|---|---|---|
| sza | -0.19 | -0.05 |
| vza | -0.07 | -0.04 |
| $\chi^2$ | 0.20 | 0.00 |
| $\chi^2_{NIR}$ | 0.18 | 0.09 |
| $\chi^2_{SWIR1}$ | 0.20 | -0.04 |
| $\chi^2_{SWIR2}$ | 0.15 | -0.05 |
| Blended albedo | 0.16 | 0.18 |
| sev | 0.07 | 0.10 |
| $\alpha^s$ | 0.04 | 0.01 |
| $\tau_{0.765}$ | -0.05 | -0.02 |
| Aerosol ratio parameter | -0.02 | -0.05 |
| water column | 0.20 | 0.06 |
| Ioff1 | -0.21 | -0.15 |
| Ioff2 | 0.05 | 0.10 |
| Ioff3 | -0.11 | -0.09 |

---

## Author Comment (AC3) · 1 May 2018

Here, we include the revised manuscript.

Please also note the supplement to this comment:
https://www.atmos-meas-tech-discuss.net/amt-2017-415/amt-2017-415-AC3-supplement.pdf

———————————————————

---

## Author Response (AR1)

We thank all reviewers for their constructive comments, which helped to improve the paper. Below, we address all comments point-by-point.

**reviewer1**
This paper describes the XCO2 retrieval from OCO-2 spectra with the RemoTeC algorithm for two years of data, focused on TCCON collocations for validation purposes. OCO-2 currently delivers the most accurate and the largest dataset of NIR / SWIR radiance measurements for XCO2 estimation. RemoTeC is widely acknowledged as a state-of-the-art retrieval algorithm, already successfully applied to GOSAT. The retrieval of OCO-2 with RemoTeC is therefore widely expected and this work largely deserves a dedicated publication. The paper is well written and concise. The main properties of OCO-2 are clearly reminded. The main assumptions of RemoTeC are recalled, but I understand that a detailed description of the algorithm requires to read references, which may be a weakness for the self consistency of the paper. Maybe the paper should be more precise about the modification of the algorithm for OCO-2. The methodology is based on the systematic comparison with several TCCON stations. This is a classical, rigorous and probably the most accurate strategy for XCO2 missions, as the column sensitivities are similar and strong efforts have been made to trace the TCCON network to the WMO XCO2 standard. Such validation work requires estimation of random error, global and regional biases, which can only be obtained at a reasonable cost with the large data set of the TCCON network. The choice of a period larger than 1 year is essential to remove the seasonal effects.

The main result is that the residuals biases of OCO-2 / RemoTeC with the TCCON global network is lower than 0.1 ppm in absolute value, and up to 1 ppm when looking at individual stations. These low values, of the same order as the OCO-2 L2, prove the quality of RemoTeC and its application to OCO-2. The remaining station to station biases are still high for the needs of the flux community, meaning that research must continue to improve the retrieval scheme and the understanding of the instrument (beyond the scope of this paper). The bias correction shows its efficiency to empirically reduce the biases, but the magnitude of the correction is still too high to give a solid confidence in the final value.

I have some questions and remarks that I would like the authors to address before publication. These will probably not require new calculations, but only precisions and additional materials. I will try to focus my questions on the application of RemoTeC to OCO-2 and not to the RemoTeC algorithm itself which was the subject of previous papers properly quoted. (1)-One of the drawbacks of this paper is that literature on this topic is large, and the reasons of some assumptions have now become implicit (and could sometimes deserve to be questioned once again). (2)-I also noticed that several results are given only in the text whereas they show be given dedicated figures or tables (see comments). This has to be corrected before publication. (3)-Finally, I was sometimes lost in the different statistics indicator (target, land, ocean; all footprints or daily averages; global bias, station to station bias, standard deviation). A clearer presentation and interpretation of them would be welcome before publication.

C1-Page 3 line 13: The objective of the study requires further justification than « to enhance the reliability and confidence of the data product ». What does this study aim at? To challenge the official OCO-2 Level 2 (L2)? To improve RemoTeC through its application to the new OCO-2 dataset, more accurate than GOSAT? Will a new OCO-2 / RemoTeC be proposed in the future?
R1-. We added a phrase " We expect that application of RemoTeC to OCO-2 data will lead to a better understanding to the capabilities and limitations of the OCO-2 instrument and the operational level-2 data product. Furthermore, we see this work as a first step towards processing a larger data set with RemoTeC." For example, in this paper we show that RemoTeC the bias correction only has a minor effect on the XCO2 retrieval accuracy, while for the official level-2 product a much larger correction is needed. This suggests that the need for bias correction is for large part caused by the algorithm itself rather than by instrument

related errors. Also we show that it is needed to fit an intensity offset which gives insight into instrumental errors of the OCO-2 instrument.

C2-As already mentioned, there is a lack of description of the algorithm, largely given by references. This is however very important to understand the differences with the OCO-2 L2.

R2- One major modification on RemoTeC/OCO-2 is that now we adopted a vector radiative transfer model (LINTRAN V2) to the retrieval scheme. Scattering is considered for ocean glint retrievals. Before, in GOSAT application, RemoTeC uses a scalar radiative transfer model for land and performs non-scattering retrieval for ocean glint. We add " For OCO-2 application, several modifications have been made to the algorithm: (1) a vector radiative transfer model (LINTRAN V2) is employed in the retrieval scheme; (2) Aerosol scattering effects are taken into account for ocean glint retrievals; (3) Information on pressure profiles, humidity and temperature are extracted from the ECMWF data with a resolution of 0.125 by 0.125 instead of 0.75 by 0.75 previously. ".

C3- p4 l12: 5° around a TCCON station is very large (~500km). In such an area the CO2 may not be considered as uniform. What is your justification? Did you make any error budget, any sensitivity study?
R3- The CO2 may be inhomogeneous in this area. However, in the validation we do not see a clear dependency between $XCO_2$ difference and the collocation distance, as shown in Figure 1. We add "Here, the dependency of difference with collocation distance and surface pressure is negligible. " in section 4.1 in the paper.

[Figure]

Figure 1. Dependency between $XCO_2$ difference and TCCON distance (unit degree).

C4-p4 l16: Why do you restrict VZA to <30° and not to a larger value? Is there also a restriction on SZA? Table 1 gives some information but in contradiction for VZA, maybe because it only applies to land and glint? Please precise.
R4- Here, we select a subset of Target data by restricting VZA<30° only for computational reasons. We do this only for target data because the amount of data becomes too large otherwise. Current retrieved target data already include more than 200.000 soundings (before quality filtering).  The restriction on SZA (<70°) is listed in Table 1.  We add a phrase " This viewing zenith angle restriction has only been applied for target observations for time efficiency".

C5-- p5 & 6: I think the paper deserves a table describing exactly the content of the state vector.
R5- Thanks for the suggestion. Table 1 is added to describe elements of the full state vector.

C6-p6 l13: the description of the cloud screening is too light, I understand it is a copy of what is done by OCO-2. Do you use the information from the OCO-2 pre-processing, or did you develop your own algorithm? Did you make any performance study, and associated XCO2 sensitivity study? 30% is higher than the performance reached by OCO-2 (for land and ocean).
R6- It is not a copy of OCO-2 cloud filtering. We use our own no-scattering retrieval algorithm. We also do not use OCO-2 pre-processing and we use a pre-processing algorithm developed at SRON. We modified in the paper " For this purpose, we implemented a fast non-scattering retrieval as part of the RemoTeC and........ Cloud filtering are performed by applying following criteria: 0.885<O2_ret/O2_ecmwf<1.020, 0.990< CO2_swir1/CO2_swir2 <1.035 and 0.950<H2O_swir1/H2O_swir2<1.060.......For target, land and ocean glint observations, the percentage of clear soundings are 24%, 28% and 34%, respectively. For now, we mainly use those ratios as a option to filter cloud contaminated cases in the retrieval. "

C7-p6 l28: please explain the reason why you separate land and ocean evaluation. Is it based only the aerosol argument (p7 l19)? Was it decided from the OCO-2 feedback ? OCO-2 does also but with another separation between land nadir and land glint.
R7- Land and ocean evaluations are separated because they have very different sensitivity to aerosols. In contrast, we found that land nadir and land glint are very similar in terms of performance against TCCON so in our opinion there was no need to separate the two.. We now add "The separation is due to the fact that land and ocean surface reflections are modeled differently".

C8-p6 l33: the assumption that TCCON station to station variability is zero is very strong and may not be excluded when interpreting the results.
R8-Yes, that's true. We emphasize this "However, as discussed by Kulawik et al. (2016); Buchwitz et al. (2017b), individual stations have a year-to-year variability of ~ 0.3 ppm and the overall TCCON XCO2 uncertainty is around 0.4 ppm (1-sigma).".

C9-p7 l8: you talk about retrieval uncertainties; these uncertainties may be instrument dependent. Compared to Butz et al 2011, Gueret et al 2013b, did you reconsider your filters for OCO-2?
R9-Yes, the filters are reconsidered. For example. The range of aerosol parameters are filtered differently compared with that used for GOSAT retrievals[ref1]. Moreover, some specific filters for OCO-2 are used such as intensity-offset ratios in SWIR1 and SWIR2 channels.

C10-p7 l25: I don't understand why you say « we look for possible correlations of errors with instrumental, geophysical, meteorological and retrieved parameters ». Actually, here you do not look for such correlations (as would the OCO-2 Bias Correction do), you only calculate a regression with chi2, which is different. This is an original bias correction and, as far as I know, it is the first time it is applied. What made you adopt such methodology? To my mind its drawbacks are that you loose interesting spectral information about the residuals. An error in retrieved albedo may lead to a large chi2 whereas it has very limited impact on XCO2. An error in line-mixing may lead to a small chi2 residual but have a strong impact on XCO2. I clearly do not say the approach is wrong, but I think that it is new and should really deserve deep study. The shape of the chi2 spectra would deserve attempts of interpretation. Why would you have only to regress with chi2 in SWIR-1 and not in the other bands? Why would this bias correction be required only for land, not for oceans? The spectroscopy and the instrument are the same. You say in section 4.1 the aerosol contribution is weak in ocean glint measurement, that could be an explanation but aerosols are not the only source of bias.

R10- We checked the correlation between the parameters mentioned here and we see relatively high correlation with chi2. In the 'Supporting Information' (SI)  (**Figure s1**), we include the correlation plots with six parameters: air mass, water column, blended albedo, mean signal in O2 A-band, aerosol ratio and aerosol size parameter.  We only do the regression with chi2 in SWIR1 (similar performance can be achieved by using chi2 in the SWIR2 band) simply because it gives the best validation results after bias correction (although it should be mentioned that the overall effect is small). We also tried other parameters like surface albedos, mean signal in O2A band, water column and overall fit residual to do the bias correction, but the station-to-station bias becomes somewhat worse.  The correlation with chi2 tells us that the XCO2 error (before bias correction) increases when the forward model is less capable in fitting the measurements or if the pure instrumental noise becomes small (and the chi2 large if the fit residuals stay the same) .The latter effect may happen over bright surfaces where it is more difficult to account for aerosol scattering.. The main effect of the bias correction over land is reducing the overall mean bias. For ocean, we directly subtract a mean bias. For sure, aerosols are not the only source of bias, but I think it is still outstanding among possible bias sources in our retrieval.

C11-p22: figure 4 should exhibit a fit, be given for lands and oceans, and for the 3 spectral bands.
R11- A linear fit is included in Figure 4. For ocean, these parameters are not used in the bias correction and we only subtract an overall mean bias.  The term fit residual was incorrectly chosen and hence we changed it to chi2.

C12-p8 l9: you say some correlation with parameters of table 1 are reduced, you clearly have to present these correlations by a figure or table before and after bias correction. Otherwise we cannot accept such affirmation.
R12-We add **Table s1** in the 'Supporting Information' (SI) section to list correlations between parameters of Table 1, before and after bias correction.

C13-p8 l19: please show fig 2 before and after bias correction. Giving a rough value in the text (~0.1ppm) is not enough.
R13- The results before bias correction are shown in SI **Figure s3**.

C14-p8 l23: please define your averaging. I understand that in fig 2 (no averaging), you plot every OCO-2 single footprint minus the TCCON of the area at the same time. I understand that in fig 5, 6, 7 you average every OCO-2 single footprint – TCCON at the same in a window 5°*5°*2h, is that true? Is it mean(OCO-2) – mean(TCCON)? You mention a « daily averaging » p8 l32, but this term is confusing because you may encounter several collocations with several TCCON during the same day. In such a case, are the data of different TCCON in the same average? If not, you should maybe talk about « overpass averaging »?
R14-Indeed, we are actually doing overpass averaging and compare mean(OCO-2) with mean(TCCON). We modify this through the paper.

C15- p8 l23: Please explain why you make an averaging in 4.2 whereas you do not in 4.1. I guess that in 4.1 you need to keep the individual parameters for your bias correction derivation, but this clearly needs to be explained. For this reason, fig 2 and fig 5,6,7 cannot be directly compared, and that is why the effect of bias correction is difficult to assess.
-p9 l1 and fig 5,6,7: Please also give the figures before bias correction for fig 5,6,7, and give the

associated standard deviation as for p8 l33.

R15- To give a better view on the effects of bias correction, we include overpass averaging figures before bias correction in the attachment (**Figures s4,s5,s6**). We add sentences to explain why we use individual retrievals in the bias correction "When comparing individual retrieval results with collocated TCCON measurements, we look for possible corrections of errors with instrumental, geophysical, meteorological and retrieved parameters. This correction should be valid for each single sounding and thus evaluated with individual results."

C16- p9 l10: please illustrate the effect of bias correction by giving figure 8 also for before bias correction.
- p9: I know you made the assumption the TCCON stations are consistent (p6 l34), but I am disappointed that you do not try to interpret the station to station variability in terms of residual bias of OCO-2 / RemoTeC or actual differences between TCCON stations. The fact that you do not see the same station to station bias in land and oceans modes could suggest there are still biases in OCO-2 /RemoTeC.

R16- We now include results before bias correction as that of Fig. 8, Fig.9 and Fig.10 in attachment (**Figure s7, s8, S9**). The station to station bias in land and ocean retrievals are close to each other(0.41 ppm vs 0.44 ppm) but it is important to note that these values are derived from different (number of) stations.

C17- p10 l6: Did you try also the same retrieval as fig 6 without the fit of the offset in the O2 band? As you mention later, there could be a link with the stronger internal reflections in this band.

R17-No, for now this retrieval setting (without the fit of the offset in the O2 band) has not been tested.

C18- p10 l12: I am not convinced by the explanation of the lack of SIF fitting, since the behaviour of the SIF and the offset is very different (SIF exhibits atmospheric absorptions).

C18-We deleted this phrase.

C19- p10 section 5: I think the comparison with the previous OCO-2 / ACOS – TCCON and with the previous GOSAT / RemoTeC – TCCON are very important. But here the discussion is poor, mentioning only the common results in terms of standard deviation. This section really deserves to compare biases (global, station to station, etc.), as it was the case for section 4. This could help understand the origin of biases (from TCCON, from the instrument, the retrieval code). This should be done before and after bias correction.

C19- We added a paragraph in section 5 that directly compares ACOS and RemoTeC before and after bias correction for common data points.

C20- p2 l9: for clarity I would make a new paragraph.

R20-modified

C21- p4 l3: please precise if you use the tabulated instrumental functions given in the OCO-2 products.

R21-modified

C22- p4 l12: you use a requirement in degrees rather than in km, why? This makes a distance criterion in km variable according to the station, which is not suitable.

R22-As can be seen above in Figure.1, there is no clear dependency between XCO2 difference and distance under current collocation criteria.

C23- p4 l18: please precise is you make your own calculation of surface pressure from ECMWF and MNT

(this information could also be read from the OCO-2 L2 data). How do you interpolate ECMWF and how to select SRTM grid points? This question makes sense since you do not retrieve surface pressure in your state vector.

R23- The interpolation is performed with linear interpolation in time and nearest neighbor in space. All SRTM grid points within certain footprint are used to get its elevation and variation. Explanations are now added' The interpolation is performed with linear interpolation in time and nearest neighbor in space. … For each OCO-2 footprint, all SRTM grid points within the boundary are collected to get mean surface elevation and its variation. '.

C24- p4 l23: Your initial guess for CO2 and CH4 comes from different years, therefore it is subject to inter-annual variability. What is the sensitivity of your retrieval to this first guess?

R24-The annual variability of prior CO2 column is not considered but the effect of the prior is very small so we are convinced this is not a problem.

[Figure]

Figure 2. Dependency between $XCO_2$ difference and prior variation (unit ppm).

ref1: Butz, A., Guerlet, S., Hasekamp, O., Schepers, D., Galli, A., Aben, I., Frankenberg, C., Hartmann, J., Tran, H., Kuze, A., Keppel-Aleks, G., Toon, G., Wunch, D., Wennberg, P., Deutscher, N., Griffith, D., Macatangay, R., Messerschmidt, J., Notholt, J. and Warneke, T. (2011). Toward accurate CO2 and CH4 observations from GOSAT. Geophysical Research Letters, 38 (14), 1-6.

**reviewer2**

This paper is describes the relatively straightforward application of the RemoTeC retrieval algorithm to data from the Orbiting Carbon Observatory-2 (OCO-2). RemoTeC was written to retrieve $CO_2$ and $CH_4$ column-average concentrations from the Greenhouse Gases Observing Satellite (GOSAT), and has been improved and validated over the years, as described in a number of publications. The authors have applied this mature algorithm to OCO-2 data to retrieve $CO_2$ (OCO-2 does not have the $CH_4$ band that GOSAT possesses), and find that after a few slight modifications, the error statistics of OCO-2 retrievals vs. ground truth data compare favorably both to the operational OCO-2 product as well as to RemoTeC retrievals of CO2 from GOSAT.

C-General Comments

The paper is useful in that it shows that the RemoTeC algorithm can be successfully applied to OCO-2, though it is relatively dry and offers few new physical insights into sources of error/bias in the OCO-2 measurements. However, is it worthwhile piece of work, and I recommend publication after making some minor revisions. My only main comment on the paper has to do with the filtering and bias correction, for which the bottom-line recipes are given. Some more information would be welcome. For instance, what other parameters were investigated for bias correction or filtering, such as the 1/(size parameter) variable used in GOSAT bias correction (Guerlet et al, 2013b)? Was the $\omega_s$ parameter of Guerlet et al. (2013b) found not to be useful for OCO-2, even though it was for GOSAT? A figure similar to that of Figure 11 in Guerlet et al. (2013b) would be very useful here to see how similar/different GOSAT vs. OCO-2 retrieval biases are. Also, how stringent were your filters overall –did you filter out 10% L2-processed soundings, 50%, etc? How was this different over land and ocean? A throughput map would be useful.

R-The correlations of XCO2 difference with other parameters as used for GOSAT are now shown in Figure S1 in attachment. We tried the potential bias correction parameter (aerosol size, reff) as previously used by GOSAT retrievals. However, as can be seen in the bottom right panel of **Fig.S1** in 'Supporting Information' (SI) there is no clear dependency between this parameter.

For the overall throughput, we add in the paper "The overall L2-processed throughput is around 15%. When estimated separately, the percentages are 15.8%, 14.0% and 16.0% for target, land and ocean soundings, respectively."

C1- P2, L24: "XCO2 retrievals with this level of accuracy [<1%] can provide valuable information on...sources and sinks..." No! 1% = 4 ppm. We know that regional biases even 1 ppm (0.25%) in XCO2 are too large (Chevallier et al, 2014). Please modify or remove this statement.

R1-Modified. The statement now becomes " The XCO2 derived from GOSAT has an accuracy in the order of a few tents of a percent. XCO2 retrievals with this level of accuracy can provide valuable information on the variation of CO2 ."

C2-P3, L2: "by aerosols and cirrus." Do water clouds not have any effect on scattering? I suggest changing this statement to "by aerosols and clouds."
R2-modified.

C3-P3, 1 st paragraph. The authors describe a number of XCO2 retrieval algorithms but this list is certainly not exhaustive. There are the BESD and FOCAL retrievals from M. Reuter, the TanSAT retrieval from D. Yang, and various versions of the PPDF retrieval of Oshchepkov and Bril. You should either cite these or make clear that you are not exhaustively listing all available retrievals.

R3- Indeed, we are not trying to list all available retrieval algorithms so we use the word 'including'.

C4- P4, L22: "barometric law". Do you not mean the hypsometric equation (which combines the ideal gas law with the hydrostatic equation). They may be equivalent, I'm not entirely sure. But usually in this context, it is referred to as the hypsometric equation.
R4- Modified.

P4, L23: Are your priors adjusted for the secular growth rate of CO2 (since you just say you use CT from 2013)? Seems like you should, or you could probably introduce an artificial trend in your retrievals.
R5- No, for now, the growth rate is not considered in the prior. It is important to note that the retrieval results are hardly affected by the prior, see Figure 2.

P5, L20: Does Sy include any estimate of forward model error, as you previously implied it might (P5, L2). Similar are the noise estimates taken from the OCO-2 suggested formulation, or do you calculate your own noise estimates somehow?
R6-Sy only include noise estimates from the OCO-2 suggested  formulation. Modified to be clear.

P6, L9-10: Please discuss whether the per-band radiance offsets were needed for GOSAT. My understanding is that they were needed for band 1, but not the other two bands. You could instead bring this up in section 4.3 as well, but I think it's important to contrast this need for the offsets in OCO-2 vs. that of GOSAT. For instance – I was thinking that maybe you needed them because your retrieval doesn't explicitly retrieve cirrus, so it would be difficult to retrieval soundings with a cirrus layer overlying a thin aerosol layer, which is a pretty common situation. Retrieval the 3 per-band offsets would be a pretty easy way to fake it. But that would likely then also be needed for GOSAT. Some discussion on this would be useful.
R7- For GOSAT the intensity offset fitting is only needed for the O2A band but not for the SWIR bands.  This suggests that the offset corrects for an instrumental artefact rather than a retrieval artefact. The state vector differences between OCO-2 and GOSAT are now added in 4.3 section.

P7, L14: $\chi^2$ is the symbol usually referred to as the total chi-squared. What you show is much more similar to the "reduced chi-squared", which is the total chi- squared divided by the number of degrees of freedom (# channels - # retrieved parameters). You really are giving the mean chi-squared per channel. You should make this clear, and that a value around unity would indicate a fit that is in line with the noise. Values consistently higher than unity mean there are the systematic errors in the forward model that are not able to be fitted away.
R8- Indeed, $\chi^2$ is the "reduced chi-squared ". Modified accordingly in the paper.

P8, top: In the discussion of using the SWIR-1 chi-squared as a bias correction parameter, it would be nice to lengthen this discussion. Does SWIR-2 chi-squared perform similarly? Other parameters? Mention that r=0.2 means that 0.04 percent of the variance is explained (or it will reduce the standard deviation by about 2%). Why do you include the offset "d" parameter when you already include a global bias correction? They would be directly related to each other. Does this multiplicative formula (equation 4) work better than an additive equation? Finally, it would be valuable if you could speculate on why this parameter seems to be correlated with the bias over land. And perhaps on why it is NOT correlated with the bias over water. Are the chi-squared values much lower over water? Finally, what is the spatial distribution of this parameter? Is it highly scattered or does it seem to remove coherent regional biases?

R9- For bias correction, using SWIR-2 chi-squared gives similar results as using SWIR-1 chi-squared when we look at some overall statistics like station-to-station bias, or standard deviation. Using other parameters like those listed in Figure S1, the performance is different and can increase the station-to-station bias. Indeed, the goal of "d" in the bias correction is to correct a global bias. We tried to keep the bias correction purely multiplicative, since the leading scaling term would just link the spectroscopic calibration to the in-situ calibration. The performance is more or less the same with an additive equation.

The reason why the SWIR-1 chi-squared is highly correlated with the bias over land but not with that over ocean is probably related to the fact that high chi2 values over land are often related to bright surfaces. We know that retrieving aerosol over bright land surface is challenging. What we also see here is that SWIR-1 chi-squared is highly correlated (cor is around 0.75) with land surface albedo. However, using albedo directly to do the bias correction can NOT achieve similar performance and will make some statistics worse, for example seasonal variations. So, apart from bright surfaces, the bias correction with chi2 corrects XCO2 retrievals for cases where the forward model is less capable of fitting the measurements. By doing the correction, as we can see in the paper, we can reduce regional biases since the station-to-staion bias become less. For ocean glint, as we mentioned, aerosols play a less important role and mainly act as an extinction layer. Thus, we cannot see similar feature with land retrievals.

P9, L10: Some comments on why the effect of the bias correction is largest for those 3 stations would be welcome. It seems like it should be substantial for all over-land stations, unless the chi-squared values were just worse for those stations. My guess is that your chi-squared is going to be correlated with SNR or surface albedo, and brighter surfaces will have larger corrections. If you plotted the mean correction on a map, this would probably become obvious.
R10- Indeed, the chi-squared is highly correlated (correlation coefficient is around 0.75) with surface albedo. This could be partly attributed to aerosols since it is difficult to account scattering effects of aerosols over bright surfaces. We added a comment with the reason for the large correction effect for the 3 stations, " This happens due to that the goodness of fit is highly correlated with surface albedo and thus make the corrections apparently to regions with large albedos.".

Section 4.3 As mentioned before, contrast with the offset approach for GOSAT. Is the behavior of the fitted radiance offsets similar over land and ocean? How correlated are the fitted offsets for the 3 bands? (either in absolute terms, or relative to the mean radiance in their respective bands) If they are highly correlated, or not, that would give you a clue what they are correcting for (either cirrus, as I hypothesized earlier, or some instrument effect that is particular to OCO-2, and perhaps not GOSAT).
R11- The fitted radiance offsets are similar in retrievals over land and ocean. The intensity offsets for the 3 bands are moderately correlated with each other (around 0.35).

P11, top: Are the GOSAT vs. OCO-2 error statistics vs. TCCON similar for both land and ocean soundings?
R12- It is difficult to compare OCO-2 ocean retrievals to that of GOSAT because the collocated TCCON sites are quite limited (4 stations).

Figures: in many of the figures, the font sizes make reading some of the text difficult (axis labels, bias numbers , TCCON site names, etc). Please try to make them bigger to increase legibility.

R13-Thanks for the suggestion. We updated this in the revised manuscript.

Technical comments
P4, various: spectral samplings à spectral samples
P5, L10: "radiative transfer model Hasekamp…" à "radiative transfer model
(Hasekamp…"
P10, L8: proportional mis-spelled
R14- modified.

[revised manuscript text omitted]